# BayesDAG: Gradient-Based Posterior Inference for Causal Discovery

**Yashas Annadani**[†*1,3,4]    **Nick Pawlowski**[2]    **Joel Jennings**[2]    **Stefan Bauer**[3,4]

**Cheng Zhang**[2]    **Wenbo Gong**[*2]

[1] KTH Royal Institute of Technology, Stockholm    [2] Microsoft Research

[3] Helmholtz AI, Munich [4] TU Munich

## Abstract

Bayesian causal discovery aims to infer the posterior distribution over causal models from observed data, quantifying epistemic uncertainty and benefiting downstream tasks. However, computational challenges arise due to joint inference over combinatorial space of Directed Acyclic Graphs (DAGs) and nonlinear functions. Despite recent progress towards efficient posterior inference over DAGs, existing methods are either limited to variational inference on node permutation matrices for linear causal models, leading to compromised inference accuracy, or continuous relaxation of adjacency matrices constrained by a DAG regularizer, which cannot ensure resulting graphs are DAGs. In this work, we introduce a scalable Bayesian causal discovery framework based on a combination of stochastic gradient Markov Chain Monte Carlo (SG-MCMC) and Variational Inference (VI) that overcomes these limitations. Our approach directly samples DAGs from the posterior without requiring any DAG regularization, simultaneously draws function parameter samples and is applicable to both linear and nonlinear causal models. To enable our approach, we derive a novel equivalence to the permutation-based DAG learning, which opens up possibilities of using any relaxed gradient estimator defined over permutations. To our knowledge, this is the first framework applying gradient-based MCMC sampling for causal discovery. Empirical evaluation on synthetic and real-world datasets demonstrate our approach's effectiveness compared to state-of-the-art baselines.

## 1 Introduction

The quest for discovering causal relationships in data-generating processes lies at the heart of empirical sciences and decision-making [56, 59, 68]. Structural Causal Models (SCMs) [52] and their associated Directed Acyclic Graphs (DAGs) provide a robust mathematical framework for modeling such relationships. Knowledge of the underlying SCM and its corresponding DAG permits predictions of unseen interventions and causal reasoning, thus making causal discovery – learning an unknown SCM and its associated DAG from observed data – a subject of extensive research [54, 60].

In contrast to traditional methods that infer a single graph or its Markov equivalence class (MEC) [14, 60], Bayesian causal discovery [21, 33, 65] aims to infer a posterior distribution over SCMs and their DAGs from observed data. This approach encapsulates the epistemic uncertainty, degree of confidence in every causal hypothesis, which is particularly valuable for real-world applications when data is scarce. It is also beneficial for downstream tasks such as experimental design [2, 4, 49, 64].

---

[*]Equal contribution.   [†] Work done during internship at Microsoft Research.   Correspondence to wenbogong@microsoft.com

37th Conference on Neural Information Processing Systems (NeurIPS 2023).

The central challenge in Bayesian causal discovery lies in inferring the posterior distribution over the union of the exponentially growing (discrete) DAGs and (continuous) function parameters. Prior works have used Markov Chain Monte Carlo (MCMC) to directly sample DAGs or bootstrap traditional discovery methods [14, 49, 65], but these methods are typically limited to linear models which admit closed-form marginalization over continuous parameters. Recent advances have begun to utilize gradient information for more efficient inference. These approaches are either: (1) DAG regularizer-based methods, e.g. DIBS [43], which use continuous relaxation of adjacency matrices together with DAG regularizer [77]. But DIBS formulation fails to model edge co-dependencies and suffer from inference quality due to its inference engine (Stein variational gradient descent) [27, 29]. Additionally, all DAG regularizer based methods cannot guarantee DAG generation; (2) permutation-based DAG learning, which directly infers permutation matrices and guarantees to generate DAGs. However, existing works focus on using only variational inference [11, 16], which may suffer from inaccurate inference quality [28, 61, 66] and is sometimes restricted to only linear models [16].

In this work, we introduce `BayesDAG`, a gradient-based Bayesian causal discovery framework that overcomes the above limitations. Our contributions are:

1. We prove that an augmented space of edge beliefs and node potentials $(\boldsymbol{W}, \boldsymbol{p})$, similar to NoCurl [72], permits equivalent Bayesian inference in DAG space without the need for any regularizer. (Section 3.1)

2. We derive an equivalence relation from this augmented space to permutation-based DAG learning which provides a general framework for gradient-based posterior inference. (Section 3.2)

3. Based on this general framework, we propose a scalable Bayesian causal discovery that is model-agnostic for linear and non-linear cases and also offers improved inference quality. We instantiate our approach through two formulations: (1) a combination of SG-MCMC and VI (2) SG-MCMC with a continuous relaxation. (Section 4)

4. We demonstrate the effectiveness of our approach in providing accurate Bayesian inference quality and superior causal discovery performance with comprehensive empirical evaluations on various datasets. We also demonstrate that our method can be easily scaled to 100 variables with nonlinear relationships. (Section 6)

## 2 Background

**Causal Graph and Structural Causal Model**    Consider a data generation process with $d$ variables $\boldsymbol{X} \in \mathbb{R}^d$. The causal relationships among these variables is represented by a Structural Causal Model (SCM) which consists of a set of structural equations [54] where each variable $X_i$ is a function of its direct causes $\boldsymbol{X}_{\mathbf{Pa}^i}$ and an exogenous noise variable $\epsilon_i$ with distribution $P_{\epsilon_i}$:

$$X_i := f_i(\boldsymbol{X}_{\mathbf{Pa}^i}, \epsilon_i) \tag{1}$$

These equations induce a causal graph $\boldsymbol{G} = (\boldsymbol{V}, \boldsymbol{E})$, comprising a node set $\boldsymbol{V}$ with $|\boldsymbol{V}| = d$ indexing the variables $\boldsymbol{X}$ and a directed edge set $\boldsymbol{E}$. If a directed edge $e_{ij} \in \boldsymbol{E}$ exists between a node pair $v_i, v_j \in \boldsymbol{V}$ (i.e., $v_i \to v_j$), we say that $X_i$ causes $X_j$ or $X_i$ is the parent of $X_j$. We use the binary adjacency matrix $\boldsymbol{G} \in \{0,1\}^{d \times d}$ to represent the causal graph, where the entry $G_{ij} = 1$ denotes $v_i \to v_j$. A standard assumption in causality is that the structural assignments are acyclic and the induced causal graph is a DAG [9, 52], which we adopt in this work. We further assume that the SCM is causally sufficient i.e. all variables are measurable and exogenous noise variables $\epsilon_i$ are mutually independent. Throughout this work, we consider a special form of SCM called Gaussian additive noise model (ANM):

$$X_i := f_i(\boldsymbol{X}_{\mathbf{Pa}^i}) + \epsilon_i \quad \text{where} \quad \epsilon_i \sim \mathcal{N}(0, \sigma_i^2) \tag{2}$$

If the functions are not linear or constant in any of its arguments, the Gaussian ANM is structurally identifiable [34, 55].

**Bayesian Causal Discovery**    Given a dataset $\boldsymbol{D} = \{\boldsymbol{x}^{(1)}, \ldots, \boldsymbol{x}^{(N)}\}$ with i.i.d observations, underlying graph $\boldsymbol{G}$ and SCM parameters $\boldsymbol{\Theta}$, they induce a unique joint distribution $p(\boldsymbol{D}, \boldsymbol{\Theta}, \boldsymbol{G}) = p(\boldsymbol{D}|\boldsymbol{G}, \boldsymbol{\Theta})p(\boldsymbol{G}, \boldsymbol{\Theta})$ with the prior $p(\boldsymbol{G}, \boldsymbol{\Theta})$ and likelihood $p(\boldsymbol{D}|\boldsymbol{G}, \boldsymbol{\Theta})$ [21]. Under finite data and/or limited identifiability of SCM (e.g upto MEC), it is desirable to have accurate uncertainty

estimation for downstream decision making rather than inferring a single SCM and its graph (for e.g. with a maximum likelihood estimate). Bayesian causal discovery therefore aims to infer the posterior $p(\boldsymbol{G}, \boldsymbol{\Theta}|\boldsymbol{D}) = p(\boldsymbol{D}, \boldsymbol{\Theta}, \boldsymbol{G})/p(\boldsymbol{D})$. However, this posterior is intractable due to the super-exponential growth of the possible DAGs $\boldsymbol{G}$ [58] and continuously valued model parameters $\boldsymbol{\Theta}$ in nonlinear functions. VI [75] or SG-MCMC [24, 45] are two types of methods developed to tackle general Bayesian inference problems, but adaptations are required for Bayesian causal discovery.

**NoCurl Characterization**  Inferring causal graphs is challenging due to the DAG constraint. Previous works [22, 25, 40, 43, 71] directly infer adjacency matrix with the DAG regularizer [77]. However, it requires an annealing schedule, resulting in slow convergence, and no guarantees on generating DAGs. Recently, [72] introduced NoCurl, a novel characterization of the **weighted DAG** space. They define a potential $p_i \in \mathbb{R}$ for each node $i$, grouped as potential vector $\boldsymbol{p} \in \mathbb{R}^d$. Further, a gradient operator on $\boldsymbol{p}$ mapping it to a skew-symmetric matrix is introduced:

$$(\operatorname{grad} \boldsymbol{p})(i, j) = p_i - p_j \tag{3}$$

Based on the above operation, a mapping that directly maps from the augmented space $(\boldsymbol{W}, \boldsymbol{p})$ to the DAG space $\gamma(\cdot, \cdot) : \mathbb{R}^{d \times d} \times \mathbb{R}^d \to \mathbb{R}^{d \times d}$ was proposed:

$$\gamma(\boldsymbol{W}, \boldsymbol{p}) = \boldsymbol{W} \odot \operatorname{ReLU}(\operatorname{grad} \boldsymbol{p}) \tag{4}$$

where $\operatorname{ReLU}(\cdot)$ is the ReLU activation function and $\boldsymbol{W}$ is a skew-symmetric **continuously weighted** matrix. This formulation is complete (Theorem 2.1 in [72]), as any continuously weighted DAG can be represented by a $(\boldsymbol{W}, \boldsymbol{p})$ pair and vice versa. NoCurl translates the learning of a single weighted DAG to a corresponding $(\boldsymbol{W}, \boldsymbol{p})$ pair. However, direct gradient-based optimization is challenging due to a highly non-convex loss landscape, which leads to the reported failure in [72].

Although NoCurl appears suitable for our purpose, the failure in directly learning suggests non-trivial optimizations. We hypothesize that this arises from the continuously weighted matrix $\boldsymbol{W}$. In the following, we introduce our proposed parametrization inspired by NoCurl to characterize the **binary** DAG adjacency matrix.

## 3  Sampling the DAGs

In this section, we focus on the Bayesian inference over binary DAGs through a novel mapping, $\tau(\boldsymbol{W}, \boldsymbol{p})$, a modification of NoCurl. We establish the validity of performing Bayesian inference within $(\boldsymbol{W}, \boldsymbol{p})$ space utilizing $\tau$ (Section 3.1). However, $\tau$ yields uninformative gradient during backpropagation, a challenge we overcome by deriving an equivalent formulation based on permutation-based DAG learning, thereby enabling the use of relaxed gradient estimators (Section 3.2).

### 3.1  Bayesian Inference in $W, p$ Space

The NoCurl formulation (Equation (4)) focuses on learning *a single weighted* DAG, which is not directly useful for our purpose. We need to address two key questions: (1) considering only binary adjacency matrices without weights; (2) ensuring Bayesian inference in $(\boldsymbol{W}, \boldsymbol{p})$ is valid.

We note that the proposed transformation in NoCurl $\gamma$ (Equation (4) can be hard to optimize for the following reasons: (i) $\operatorname{ReLU}(\operatorname{grad} \boldsymbol{p})$ gives a fully connected DAG. The main purpose of $\boldsymbol{W}$ matrix therefore is to disable the edges. Continuous $\boldsymbol{W}$ requires thresholding to properly disable the edges, since it is hard for a continuous matrix to learn exactly 0 during the optimization; (ii) $\operatorname{ReLU}(\operatorname{grad} \boldsymbol{p})$ and $\boldsymbol{W}$ are both continuous valued matrices. Thus, learning of the edge weights and DAG structure are not explicitly separated, resulting in complicated non-convex optimizations[2]. Parameterizing the search space in terms of binary adjacency matrices significantly simplifies the optimization complexity as the aforementioned issues are circumvented. Therefore, we introduce a modification $\tau : \{0, 1\}^{d \times d} \times \mathbb{R}^d \to \{0, 1\}^{d \times d}$:

$$\tau(\boldsymbol{W}, \boldsymbol{p}) = \boldsymbol{W} \odot \operatorname{Step}(\operatorname{grad} \boldsymbol{p}) \tag{5}$$

where we abuse the term $\boldsymbol{W}$ for binary matrices, and replace $\operatorname{ReLU}(\cdot)$ with $\operatorname{Step}(\cdot)$. $\boldsymbol{W}$ acts as mask to disable the edge existence. Thus, due to the $\operatorname{Step}$, $\tau$ can only output a binary adjacency matrix.

---

[2]See discussion below Eq. 3 in [72] for more details.

Next, we show that performing Bayesian inference in such augmented $(\boldsymbol{W}, \boldsymbol{p})$ space is valid, i.e., using the posterior $p(\boldsymbol{W}, \boldsymbol{p}|\boldsymbol{D})$ to replace $p(\boldsymbol{G}|\boldsymbol{D})$. This differs from NoCurl, which focuses on a single graph rather than the validity for Bayesian inference, requiring a new theory for soundness.

**Theorem 3.1** (Equivalence of inference in $(\boldsymbol{W}, \boldsymbol{p})$ and binary DAG space). *Assume graph $\boldsymbol{G}$ is a binary adjacency matrix representing a DAG and node potential $\boldsymbol{p}$ does not contain the same values, i.e. $p_i \neq p_j \; \forall i, j$. Then, with the induced joint observational distribution $p(\boldsymbol{D}, \boldsymbol{G})$, dataset $\boldsymbol{D}$, and a corresponding prior $p(\boldsymbol{G})$, we have*

$$p(\boldsymbol{G}|\boldsymbol{D}) = \int p_\tau(\boldsymbol{p}, \boldsymbol{W}|\boldsymbol{D}) \, \mathbb{1}(\boldsymbol{G} = \tau(\boldsymbol{W}, \boldsymbol{p})) d\boldsymbol{W} d\boldsymbol{p} \tag{6}$$

*if $p(\boldsymbol{G}) = \int p_\tau(\boldsymbol{p}, \boldsymbol{W}) \, \mathbb{1}(\boldsymbol{G} = \tau(\boldsymbol{W}, \boldsymbol{p})) d\boldsymbol{W} d\boldsymbol{p}$, where $p_\tau(\boldsymbol{W}, \boldsymbol{p})$ is the prior, $\mathbb{1}(\cdot)$ is the indicator function, and $p_\tau(\boldsymbol{p}, \boldsymbol{W}|D)$ is the posterior distribution over $\boldsymbol{p}, \boldsymbol{W}$.*

Refer to Appendix B.1 for detailed proof.

This theorem guarantees that instead of performing inference directly in the constrained space (i.e. DAG space), we can apply Bayesian inference in a less complex $(\boldsymbol{W}, \boldsymbol{p})$ space where $\boldsymbol{W} \in \{0, 1\}^{d \times d}$ and $\boldsymbol{p} \in \mathbb{R}^d$ without explicit constraints.

For inference of $\boldsymbol{p}$, we adopt a sampling-based approach, which is asymptotically accurate [45]. In particular, we consider SG-MCMC (refer to Section 4), which avoids the expensive Metropolis-Hastings acceptance step and scales to large datasets. We emphasize that any other suitable sampling algorithms can be directly plugged in, thanks to the generality of the framework.

However, the mapping $\tau$ does not provide meaningful gradient information for $\boldsymbol{p}$ due to the piecewise constant $\text{Step}(\cdot)$ function, which is required by SG-MCMC.

### 3.2 Equivalent Formulation

In this section, we address the above issue by deriving an equivalence to a permutation learning problem. This alternative formulation enables various techniques that can approximate the gradient of $\boldsymbol{p}$.

**Intuition** The node potential $\boldsymbol{p}$ implicitly defines a topological ordering through the mapping $\text{Step}(\text{grad}(\cdot))$. In particular, $\text{grad}(\cdot)$ outputs a skew-symmetric adjacency matrix, where each entry specifies the potential difference between nodes. $\text{Step}(\text{grad}(\cdot))$ zeros out the negative potential differences (i.e. $p_i \leq p_j$), and only permits the edge direction from higher potential to the lower one (i.e. $p_i > p_j$). This implicitly defines a sorting operation based on the descending node potentials, which can be cast as a particular $\arg\max$ problem [8, 37, 47, 50, 74] involving a permutation matrix.

**Alternative formulation** We define $\boldsymbol{L} \in \{0, 1\}^{d \times d}$ as a matrix with lower triangular part to be 1, and vector $\boldsymbol{o} = [1, \ldots, d]$. We propose the following formulation:

$$\boldsymbol{G} = \boldsymbol{W} \odot \left[ \boldsymbol{\sigma}(\boldsymbol{p}) \boldsymbol{L} \boldsymbol{\sigma}(\boldsymbol{p})^T \right] \qquad \text{where } \boldsymbol{\sigma}(\boldsymbol{p}) = \underset{\boldsymbol{\sigma}' \in \boldsymbol{\Sigma}_d}{\arg\max} \, \boldsymbol{p}^T (\boldsymbol{\sigma}' \boldsymbol{o}) \tag{7}$$

Here, $\boldsymbol{\Sigma}_d$ represents the space of all $d$ dimensional permutation matrices. The following theorem states the equivalence of this formulation to Equation (5).

**Theorem 3.2** (Equivalence to NoCurl formulation). *Assuming the conditions in Theorem 3.1 are satisfied. Then, for a given $(\boldsymbol{W}, \boldsymbol{p})$, we have*

$$\boldsymbol{G} = \boldsymbol{W} \odot \text{Step}(\text{grad} \, \boldsymbol{p}) = \boldsymbol{W} \odot \left[ \boldsymbol{\sigma}(\boldsymbol{p}) \boldsymbol{L} \boldsymbol{\sigma}(\boldsymbol{p})^T \right]$$

*where $\boldsymbol{G}$ is a DAG and $\boldsymbol{\sigma}(\boldsymbol{p})$ is defined in Equation (7).*

Refer to Appendix B.2 for details.

This theorem translates our proposed operator $\text{Step}(\text{grad}(\boldsymbol{p}))$ into finding a corresponding permutation matrix $\boldsymbol{\sigma}(\boldsymbol{p})$. Although this does not directly solve the uninformative gradient, it opens the door for approximating this gradient with the tools from the differentiable permutation literature [8, 47, 50]. For simplicity, we adopt the Sinkhorn approach [47], but we emphasize that this equivalence is general enough that any past or future approximation methods can be easily applied.

**Sinkhorn operator**   The Sinkhorn operator $\mathcal{S}(\boldsymbol{M})$ on a matrix $\boldsymbol{M}$ [1] is defined as a sequence of row and column normalizations, each is called Sinkhorn iteration.

[47] showed that the non-differentiable $\arg\max$ problem

$$\boldsymbol{\sigma} = \arg\max_{\boldsymbol{\sigma}' \in \Sigma_d} \langle \boldsymbol{\sigma}', \boldsymbol{M} \rangle \tag{8}$$

can be relaxed through an entropy regularizer with its solution being expressed by $\mathcal{S}(\cdot)$. In particular, they showed that $\mathcal{S}(\boldsymbol{M}/t) = \arg\max_{\boldsymbol{\sigma}' \in \mathcal{B}_d} \langle \boldsymbol{\sigma}', \boldsymbol{M} \rangle + th(\boldsymbol{\sigma}')$, where $h(\cdot)$ is the entropy function. This regularized solution converges to the solution of Equation (8) when $t \to 0$, i.e. $\lim_{t\to0} \mathcal{S}(\boldsymbol{M}/t)$. Since the Sinkhorn operator is differentiable, $\mathcal{S}(\boldsymbol{M}/t)$ can be viewed as a differentiable approximation to Equation (8), which can be used to obtain the solution of Equation (7). Specifically, we have

$$\arg\max_{\boldsymbol{\sigma}' \in \boldsymbol{\Sigma}_d} \boldsymbol{p}^T(\boldsymbol{\sigma}'\boldsymbol{o}) = \arg\max_{\boldsymbol{\sigma}' \in \boldsymbol{\Sigma}_d} \langle \boldsymbol{\sigma}', \boldsymbol{p}\boldsymbol{o}^T \rangle = \lim_{t\to0} \mathcal{S}(\frac{\boldsymbol{p}\boldsymbol{o}^T}{t}) \tag{9}$$

In practice, we approximate it wth $t > 0$, resulting in a doubly stochastic matrix. To get the binary permutation matrix, we apply the Hungarian algorithm [48]. During the backward pass, we use a straight-through estimator [7] for $\boldsymbol{p}$.

Some of the previous works [11, 16] have leveraged the Sinkhorn operator to model variational distributions over permutation matrices. However, they start with a full rank $\boldsymbol{M}$, which has been reported to require over **1000** Sinkho rn iterations to converge [16]. However, our formulation, based on explicit node potential $\boldsymbol{p}\boldsymbol{o}^T$, generates a rank-1 matrix, requiring much fewer Sinkhorn steps (around **300**) in practice, saving two-thirds of the computational cost.

## 4   Bayesian Causal Discovery via Sampling

In this section, we delve into two specific methodologies that are derived from the proposed framework. The first one, which will be our main focus, combines SG-MCMC and VI in a Gibbs sampling manner. The second one, which is based entirely on SG-MCMC with continuous relaxation, is also derived, but we include its details in Appendix A due to its inferior empirical performance.

### 4.1   Model Formulation

We build upon the model formulation of [22], which combines the additive noise model with neural networks to describe the functional relationship. Specifically, $X_i := f_i(\boldsymbol{X}_{\mathbf{Pa}^i}) + \epsilon_i$, where $f_i$ adheres to the adjacency relation specified by $\boldsymbol{G}$, i.e. $\partial f_i(\boldsymbol{x})/\partial x_j = 0$ if no edge exists between nodes $i$ and $j$. We define $f_i$ as

$$f_i(\boldsymbol{x}) = \zeta_i \left( \sum_{j=1}^{d} G_{ji} l_j(x_j) \right), \tag{10}$$

where $\zeta_i$ and $l_i$ are neural networks with parameters $\boldsymbol{\Theta}$, and $\boldsymbol{G}$ serves as a mask disabling non-parent values. To reduce the number of neural networks, we adopt a weight-sharing mechanism: $\zeta_i(\cdot) = \zeta(\boldsymbol{u}_i, \cdot)$ and $l_i(\cdot) = l(\boldsymbol{u}_i, \cdot)$, with trainable node embeddings $\boldsymbol{u}_i$.

**Likelihood of SCM**   The likelihood can be evaluated through the noise $\boldsymbol{\epsilon} = \boldsymbol{x} - \boldsymbol{f}(\boldsymbol{x}; \boldsymbol{\Theta})$. [22] showed that if $\boldsymbol{G}$ is a DAG, then the mapping from $\boldsymbol{\epsilon}$ to $\boldsymbol{x}$ is invertible with a Jacobian determinant of 1. Thus, the observational data likelihood is:

$$p(\boldsymbol{x}|\boldsymbol{G}) = p_\epsilon(\boldsymbol{x} - \boldsymbol{f}(\boldsymbol{x}; \boldsymbol{\Theta})) = \prod_{i=1}^{d} p_{\epsilon_i}(x_i - f_i(\boldsymbol{x}_{\mathbf{Pa}_G^i})) \tag{11}$$

**Prior design**   We implicitly define the prior $p(\boldsymbol{G})$ via $p(\boldsymbol{p}, \boldsymbol{W})$. We propose the following for the joint prior:

$$p(\boldsymbol{W}, \boldsymbol{p}, \boldsymbol{\Theta}) \propto \mathcal{N}(\boldsymbol{\Theta}; \boldsymbol{0}, \boldsymbol{I})\mathcal{N}(\boldsymbol{p}; \boldsymbol{0}, \alpha\boldsymbol{I})\mathcal{N}(\boldsymbol{W}; \boldsymbol{0}, \boldsymbol{I})\exp(-\lambda_s \|\tau(\boldsymbol{W}, \boldsymbol{p})\|_F^2)$$

where $\alpha$ controls the initialization scale of $\boldsymbol{p}$ and $\lambda_s$ controls the sparseness of $\boldsymbol{G}$.

## 4.2 Bayesian Inference of $W, p, \Theta$

The main challenge lies in the binary nature of $\boldsymbol{W} \in \{0, 1\}^{d \times d}$, which requires a discrete sampler. Although recent progress has been made [30, 62, 73, 76], these methods either involve expensive Metropolis-Hasting (MH) steps or require strong assumptions on the target posterior when handling batched gradients. To address this, we propose a combination of SG-MCMC for $\boldsymbol{p}, \Theta$ and VI for $\boldsymbol{W}$. It should be noted that our framework can incorporate any suitable discrete sampler if needed.

We employ a Gibbs sampling procedure [10], which iteratively applies (1) sampling $\boldsymbol{p}, \Theta \sim p(\boldsymbol{p}, \Theta|\boldsymbol{D}, \boldsymbol{W})$ with SG-MCMC; (2) updating the variational posterior $q_\phi(\boldsymbol{W}|\boldsymbol{p}, \boldsymbol{D}) \approx p(\boldsymbol{W}|\boldsymbol{p}, \Theta, \boldsymbol{D})$.

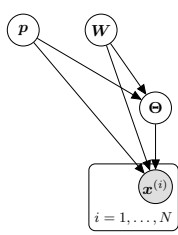

We define the posterior $p(\boldsymbol{p}, \Theta|\boldsymbol{D}, \boldsymbol{W}) \propto \exp(-U(\boldsymbol{p}, \boldsymbol{W}, \Theta))$, where $U(\boldsymbol{p}, \boldsymbol{W}, \Theta) = -\log p(\boldsymbol{p}, \boldsymbol{D}, \boldsymbol{W}, \Theta)$. SG-MCMC in continuous time defines a specific form of Itô diffusion that maintains the target distribution invariant [45] without the expensive computation of the MH step. We adopt the Euler-Maruyama discretization for simplicity. Other advanced discretization can be easily incorporated [12, 57].

Preconditioning techniques have been shown to accelerate SG-MCMC convergence [13, 28, 41, 69, 70]. We modify the sampler based on [28], which is inspired by Adam [35]. Detailed update equations can be found in Appendix C.

Figure 1: Graphical model of the inference problem.

The following proposition specifies the gradients required by SG-MCMC: $\nabla_{\boldsymbol{p}, \Theta} U(\boldsymbol{p}, \boldsymbol{W}, \Theta)$.

**Proposition 4.1.** *Assume the model is defined as above, then we have the following:*

$$\nabla_{\boldsymbol{p}} U = -\nabla_{\boldsymbol{p}} \log p(\boldsymbol{p}) - \nabla_{\boldsymbol{p}} \log p(\boldsymbol{D}|\Theta, \tau(\boldsymbol{W}, \boldsymbol{p})) \tag{12}$$

*and*

$$\nabla_\Theta U = -\nabla_\Theta \log p(\Theta) - \nabla_\Theta \log p(\boldsymbol{D}|\Theta, \tau(\boldsymbol{p}, \boldsymbol{W})) \tag{13}$$

Refer to Appendix B.5 for details.

**Variational inference for $W$**    We use the variational posterior $q_\phi(\boldsymbol{W}|\boldsymbol{p})$ to approximate the true posterior $p(\boldsymbol{W}|\boldsymbol{p}, \Theta, \boldsymbol{D})$. Specifically, we select an independent Bernoulli distribution with logits defined by the output of a neural network $\mu_\phi(\boldsymbol{p})$:

$$q_\phi(\boldsymbol{W}|\boldsymbol{p}) = \prod_{ij} Ber(\mu_\phi(\boldsymbol{p})_{ij}) \tag{14}$$

To train $q_\phi$, we derive the corresponding *evidence lower bound* (ELBO):

$$\text{ELBO}(\phi) = \mathbb{E}_{q_\phi(\boldsymbol{W}|\boldsymbol{p})} [\log p(\boldsymbol{D}, \boldsymbol{p}, \Theta|\boldsymbol{W})] - D_{\text{KL}} [q_\phi(\boldsymbol{W}|\boldsymbol{p}) \| p(\boldsymbol{W})] . \tag{15}$$

where $D_{\text{KL}}$ is the Kullback-Leibler divergence. The derivation is in Appendix B.6. Algorithm 1 summarizes this inference procedure.

**SG-MCMC with continuous relaxation**    Furthermore, we explore an alternative formulation that circumvents the need for variational inference. Instead, we employ SG-MCMC to sample $\tilde{\boldsymbol{W}}$, a continuous relaxation of $\boldsymbol{W}$, facilitating a fully sampling-based approach. For a detailed formulation, please refer to Appendix A. We report its performance in Appendix E.3, which surprisingly is inferior to SG-MCMC+VI. We hypothesize that coupling $\boldsymbol{W}, \boldsymbol{p}$ through $\mu_\phi$ is important since changes in $\boldsymbol{p}$ results in changes of the permutation matrix $\boldsymbol{\sigma}(\boldsymbol{p})$, which should also influence $\boldsymbol{W}$ accordingly during posterior inference. However, through sampling $\tilde{\boldsymbol{W}}$ with few SG-MCMC steps, this change cannot be immediately reflected, resulting in inferior performance. Thus, we focus only on the performance of SG-MCMC+VI for our experiments.

**Computational complexity**    Our proposed SG-MCMC+VI offers a notable improvement in computational cost compared to existing approaches, such as DIBS [43]. The computational complexity of

---

**Algorithm 1** `BayesDAG` SG-MCMC+VI Inference

---

**Input:** dataset $\boldsymbol{D}$; prior $p(\boldsymbol{p}, \boldsymbol{W}), p(\boldsymbol{\Theta})$; SG-MCMC sampler Sampler; sampler hyperparameters $\Psi$; network $\mu_\phi(\cdot)$; training iteration $T$.
**Output:** samples $\{\boldsymbol{\Theta}, \boldsymbol{p}\}$ and variational posterior $q_\phi$
Initialize $\boldsymbol{\Theta}^{(0)}, \boldsymbol{p}^{(0)}, \phi$
**for** $t = 1 \ldots, T$ **do**
    Sample $\boldsymbol{W}^{(t-1)} \sim q_\phi(\boldsymbol{W}|\boldsymbol{p}^{(t-1)})$
    Evaluate $\nabla_{\boldsymbol{p},\boldsymbol{\Theta}} U$ (Equations (12) and (13)) with $\boldsymbol{\Theta}^{(t-1)}, \boldsymbol{p}^{(t-1)}, \boldsymbol{W}^{(t-1)}$
    $\boldsymbol{\Theta}^{(t)}, \boldsymbol{p}^{(t)} = \text{Sampler}(\nabla_{\boldsymbol{p},\boldsymbol{\Theta}} U; \Psi)$
    **if** storing condition met **then**
        $\{\boldsymbol{p}, \boldsymbol{\Theta}\} \leftarrow \boldsymbol{p}^{(t)}, \boldsymbol{\Theta}^{(t)}$
    **end if**
    Maximize ELBO (Equation (15)) w.r.t. $\phi$ with $\boldsymbol{p}^{(t)}, \boldsymbol{\Theta}^{(t)}$
**end for**

---

our method is $O(BN_p + N_p d^3)$, where $B$ represents the batch size and $N_p$ is the number of parallel SG-MCMC chains. This former term stems from the forward and backward passes, and the latter comes from the Hungarian algorithm, which can be parallelized to further reduce computational cost. In comparison, DIBS has a complexity of $O(N_p^2 N + N_p d^3)$ with $N \gg B$ being the full dataset size. This is due to the kernel computation involving the entire dataset and the evaluation of the matrix exponential in the DAG regularizer [77]. As a result, our approach provides linear scalability w.r.t. $N_p$ with substantially smaller batch size $B$. Conversely, DIBS exhibits quadratic scaling in terms of $N_p$ and lacks support for mini-batch gradients.

## 5 Related Work

Bayesian causal discovery literature has primarily focused on inference in linear models with closed-form posteriors or marginalized parameters. Early works considered sampling directed acyclic graphs (DAGs) for discrete [15, 46, 33] and Gaussian random variables [21, 65] using Markov chain Monte Carlo (MCMC) in the DAG space. However, these approaches exhibit slow mixing and convergence [18, 32], often requiring restrictions on number of parents [38]. Alternative exact dynamic programming methods are limited to low-dimensional settings [36].

Recent advances in variational inference [75] have facilitated graph inference in DAG space, with gradient-based methods employing the NOTEARS DAG penalty [77].[3] samples DAGs from autoregressive adjacency matrix distributions, while [43] utilizes Stein variational approach [42] for DAGs and causal model parameters. [16] proposed a variational inference framework on node orderings using the gumbel-sinkhorn gradient estimator [47]. [17, 51] employ the GFlowNet framework [6] for inferring the DAG posterior. Most methods, except[43] are restricted to linear models, while [43] has high computational costs and lacks DAG generation guarantees compared to our method.

In contrast, *quasi-Bayesian* methods, such as DAG bootstrap [20], demonstrate competitive performance. DAG bootstrap resamples data and estimates a single DAG using PC [60], GES [14], or similar algorithms, weighting the obtained DAGs by their unnormalized posterior probabilities. Recent neural network-based works employ variational inference to learn DAG distributions and point estimates for nonlinear model parameters [11, 22].

## 6 Experiments

In this section, we aim to study empirically the following aspects: (1) posterior inference quality of `BayesDAG` as compared to the true posterior when the causal model is identifiable only upto Markov Equivalence Class (MEC); (2) posterior inference quality of `BayesDAG` in high dimensional nonlinear causal models (3) ablation studies of `BayesDAG` and (4) performance in semi-synthetic and real world applications. The experiment details are included in Appendix D.

**Baselines.** We mainly compare `BayesDAG` with the following baselines: Bootstrap GES (**BGES**) [14, 20], **BCD** Nets [16], Differentiable DAG Sampling (**DDS** [11]) and **DIBS** [43].

### 6.1 Evaluation on Synthetic Data

**Synthetic data.** We evaluate our method on synthetic data, where ground truth graphs are known. Following previous work, we generate data by randomly sampling DAGs from Erdos-Rènyi (ER) [19] or Scale-Free (SF) [5] graphs with per node degree 2 and drawing at random ground truth parameters for linear or nonlinear models. For $d = 5$, we use $N = 500$ training, while for higher dimensions, we use $N = 5000$. We assess performance on 30 random datasets for each setting.

**Metrics** For $d = 5$ linear models, we compare the approximate and true posterior over DAGs using Maximum Mean Discrepancy (MMD) and also evaluate the expected CPDAG Structural Hamming Distance (SHD). For higher-dimensional nonlinear models with intractable posterior, we compute the expected SHD ($\mathbb{E}$-

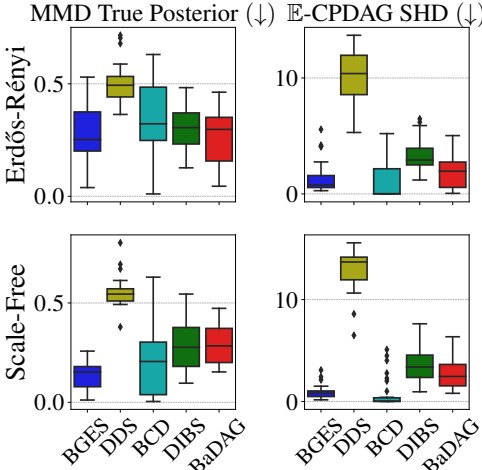

Figure 2: Posterior inference on linear synthetic datasets with $d = 5$. Metrics are computed against the true posterior. $\downarrow$ denotes lower is better.

**SHD**), expected orientation F1 score (**Edge F1**) and negative log-likelihood of the held-out data (**NLL**). Our synthetic data generation and evaluation protocol follows prior work [3, 22, 43]. All the experimental details, including how we use cross-validation to select hyperparameters is in Appendix D.

#### 6.1.1 Comparison with True Posterior

Capturing equivalence classes and quantifying epistemic uncertainty are crucial in Bayesian causal discovery. We benchmark our method against the true posterior using a 5-variable linear SCM with unequal noise variance (identifiable upto MEC [53]). The true posterior over graphs $p(\boldsymbol{G} \mid \boldsymbol{D})$ can be computed using the BGe score [23, 39]. Results in Figure 2 show that our method outperforms DIBS and DDS in both ER and SF settings. Compared to BCD, we perform better in terms of MMD in ER but worse in SF. We find that BGES performs very well in low-dimensional linear settings, but suffers significantly in more realistic nonlinear settings (see below).

#### 6.1.2 Evaluation in Higher Dimensions

We evaluate our method on high dimensional scenarios with nonlinear relations. Our approach is the first to attempt full posterior inference in nonlinear models using permutation-based methods. Results for $d = 30$ variables in Figure 3 demonstrate that BayesDAG significantly outperforms other *permutation-based approaches* and DIBS in most of the metrics. For $d = 50$, BayesDAG performs comparably to

Table 1: $\mathbb{E}$-SHD (with $95\%$ CI) for ER graphs in higher dimensional nonlinear causal models. DIBS becomes computationally prohibitive for $d > 50$.

|        | $d = 70$             | $d = 100$            |
|--------|----------------------|----------------------|
| BGES   | $355.77 \pm 18.02$   | $563.02 \pm 27.21$   |
| BCD    | $217.05 \pm 9.58$    | $362.66 \pm 29.18$   |
| DIBS   | N/A                  | N/A                  |
| BaDAG  | $\mathbf{143.70 \pm 11.61}$ | $\mathbf{295.92 \pm 24.67}$ |

DIBS in ER but a little worse in SF. However, our method achieves better NLL on held-out data compared to most baselines including DIBS for $d = 30, 50$, ER and SF settings. Only DDS gives better NLL for $d = 30$ ER setting, but this doesn't translate well to other metrics and settings. We additionally evaluate on $d \in \{70, 100\}$ variables (Table 1). We find that our method consistently outperforms the baselines with $d = 70$ and in terms of $\mathbb{E}$-SHD with $d = 100$. Full results are presented in Appendix E.2. Competitive performance for $d > 50$ in nonlinear settings further demonstrates the applicability and computational efficiency of the proposed approach. In contrast, the only fully Bayesian nonlinear method, DIBS, is not computationally efficient to run for $d > 50$.

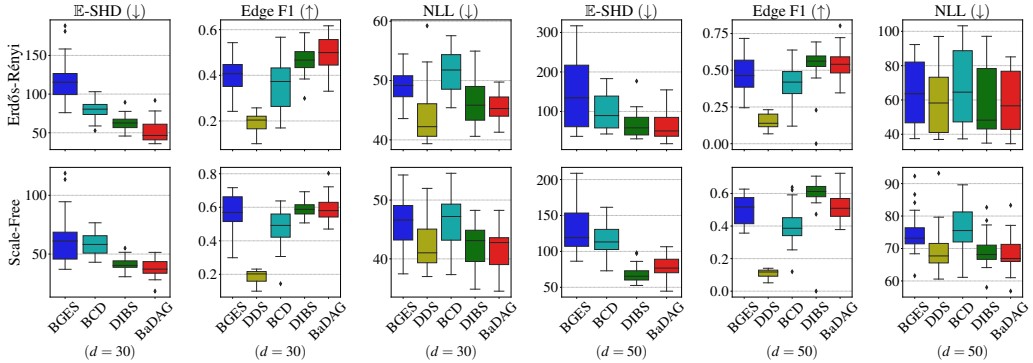

Figure 3: Posterior inference of both graph and functional parameters on synthetic datasets of nonlinear causal models with $d = 30$ and $d = 50$ variables. `BayesDAG` gives best results across most metrics and outperforms other permutation based approaches (BCD and DDS). We found DDS to perform significantly worse in terms of $\mathbb{E}$-SHD and thus has been omitted for clarity. $\downarrow$ denotes lower is better and $\uparrow$ denotes higher is better.

## 6.2 Ablation Studies

We conduct ablation studies on our method using the nonlinear ER $d = 30$ dataset.

**Initialized $p$ scale**   Figure 4a investigates the influence of the initialized scale of $p$. We found that the performance is the best with $\alpha = 0.01$ or $10^{-5}$, and deteriorates with increasing scales. This is because with larger initialization scale, the absolute value of the $p$ is large. Longer SG-MCMC updates are needed to reverse the node potential order, which hinders the exploration of possible permutations, resulting in the convergence to poor local optima.

**Number of SG-MCMC chains**   We examine the impact of the number of parallel SG-MCMC chains in Figure 4b. We observe that it does not have a significant impact on the performance, especially with respect to the $\mathbb{E}$-SHD and Edge F1 metrics.

**Injected noise level for SG-MCMC**   In Figures 4c and 4d, we study the performance differences arising from various injected noise levels for $p$ and $\Theta$ in the SG-MCMC algorithm (i.e. $s$ of the SG-MCMC formulation in Appendix C). Interestingly, the noise level of $p$ does not impact the performance as much as the level of $\Theta$. Injecting noise helps improve the performance, but a smaller noise level should be chosen for $\Theta$ to avoid divergence from optima.

## 6.3 Application 1: Evaluation on Semi-Synthetic Data

We evaluate our method on the SynTReN simulator [67]. This simulator creates synthetic transcriptional regulatory networks and produces simulated gene expression data that approximates real experimental data. We use five different simulated datasets provided by [40] with $N = 500$ samples each. Table 2 presents the results of all the methods. We find that our method recovers the true network much better in terms of $\mathbb{E}$-SHD as well as Edge F1 compared to baselines.

## 6.4 Application 2: Evaluation on Real Data

We also evaluate on a real dataset which measures the expression level of different proteins and phospholipids in human cells (called the Sachs Protein Cells Dataset) [59]. The data corresponds to a network of protein-protein interactions of 11 different proteins with 17 edges in total among them. There are 853 observational samples in total, from which we bootstrap 800 samples of 5 different datasets. It is to be noted that this data does not necessarily adhere to the additive noise and DAG assumptions, thereby having significant model misspecification. Results in Table 2 demonstrate that our method performs well as compared to the baselines even with model misspecification, proving the suitability of the proposed framework for real-world settings.

Table 2: Results (with 95% confidence intervals) on Syntren (semi-synthetic) and Sachs Protein Cells (real-world) datasets. For Syntren, results are averaged over 5 different datasets. For Sachs, results are averaged over 5 different restarts. ↓ denotes lower is better and ↑ denotes higher is better.

| | Syntren ($d = 20$) | | Sachs Protein Cells ($d = 11$) | |
|---|---|---|---|---|
| | $\mathbb{E}$-SHD ($\downarrow$) | Edge F1 ($\uparrow$) | $\mathbb{E}$-SHD ($\downarrow$) | Edge F1 ($\uparrow$) |
| BGES | $66.18 \pm 9.47$ | $\mathbf{0.21 \pm 0.05}$ | $\mathbf{16.61 \pm 0.44}$ | $0.22 \pm 0.02$ |
| DDS | $134.37 \pm 4.58$ | $0.13 \pm 0.02$ | $34.90 \pm 0.73$ | $0.21 \pm 0.02$ |
| BCD | $38.38 \pm 7.12$ | $0.15 \pm 0.07$ | $17.05 \pm 1.93$ | $0.20 \pm 0.08$ |
| DIBS | $46.43 \pm 4.12$ | $0.16 \pm 0.02$ | $22.3 \pm 0.31$ | $0.20 \pm 0.01$ |
| BaDAG | $\mathbf{34.21 \pm 2.82}$ | $\mathbf{0.20 \pm 0.02}$ | $18.92 \pm 1.0$ | $\mathbf{0.26 \pm 0.04}$ |

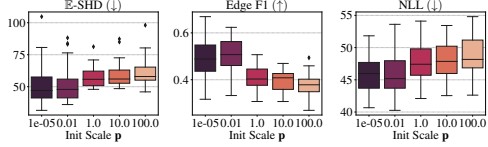

(a) Posterior inference with different initialized $\boldsymbol{p}$ scale.

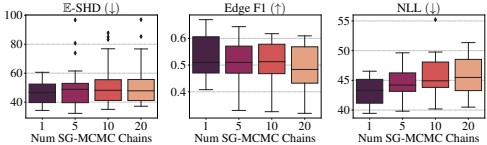

(b) Posterior inference with different number of parallel SG-MCMC chains.

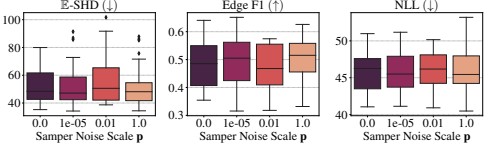

(c) Posterior inference with different level of injected noise scale for $\boldsymbol{p}$.

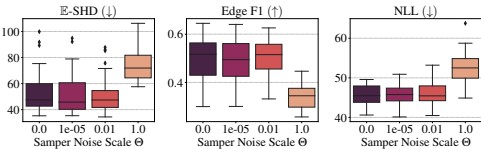

(d) Posterior inference with different level of injected noise scale for $\boldsymbol{\Theta}$.

Figure 4: Ablation study of posterior inference quality of `BayesDAG` on $d = 30$ ER synthetic dataset.

## 7 Discussion

In this work, we propose `BayesDAG`, a novel, scalable Bayesian causal discovery framework that employs SG-MCMC (and VI) to infer causal models. We establish the validity of performing Bayesian inference in the augmented $(\boldsymbol{W}, \boldsymbol{p})$ space and demonstrate its connection to permutation-based DAG learning. Furthermore, we provide two instantiations of the proposed framework that offers direct DAG sampling and model-agnosticism to linear and nonlinear relations. We demonstrate superior inference accuracy and scalability on various datasets. Future work can address some limitations: (1) designing better variational networks $\mu_\phi$ to capture the complex distributions of $\boldsymbol{W}$ compared to the simple independent Bernoulli distribution; (2) improving the performance of SG-MCMC with continuous relaxation (Appendix A), which currently does not align with its theoretical advantages compared to the SG-MCMC+VI counterpart.

**Acknowledgements.** The authors would like to thank Colleen Tyler, Maria Defante, and Lisa Parks for conversations on real-world use cases that inspired this work. YA and SB are thankful for the Swedish National Computing's Berzelius cluster for providing resources that were helpful in running some of the baselines of the paper. In addition, the authors would like to thank the anonymous reviewers for their feedback.

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

# Appendix – BayesDAG: Gradient-Based Posterior Inference for Causal Discovery

## A Joint Inference with SG-MCMC

In this section, we propose an alternative formulation that enables a joint inference framework for $\boldsymbol{p}, \boldsymbol{W}, \boldsymbol{\Theta}$ using SG-MCMC, thereby avoiding the need for variational inference for $\boldsymbol{W}$.

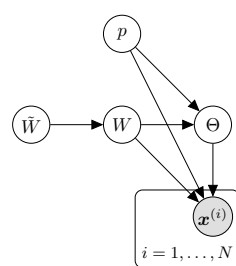

We adopt a continuous relaxation of $\boldsymbol{W}$, similar to [43], by introducing a latent variable $\tilde{\boldsymbol{W}}$. The graphical model is illustrated in Figure 5. We can define

$$p(\boldsymbol{W}|\tilde{\boldsymbol{W}}) = \prod_{i,j} p(W_{ij}|\tilde{W}_{ij}) \qquad (16)$$

with $p(W_{ij} = 1|\tilde{W}_{ij}) = \sigma(\tilde{W}_{ij})$ where $\sigma(\cdot)$ is the sigmoid function. In other words, $\tilde{W}ij$ defines the existence logits of $Wij$.

Figure 5: Graphical model with latent variable $\tilde{\boldsymbol{W}}$.

With the introduction of $\tilde{\boldsymbol{W}}$, the original posterior expectations of $p(\boldsymbol{p}, \boldsymbol{W}, \boldsymbol{\Theta}|\boldsymbol{D})$, e.g. during evaluation, can be translated using the following proposition.

**Proposition A.1** (Equivalence of posterior expectation). *Under the generative model Figure 5, we have*

$$\mathbb{E}_{p(\boldsymbol{p}, \boldsymbol{W}, \boldsymbol{\Theta}|\boldsymbol{D})}\left[\boldsymbol{f}(\boldsymbol{G} = \tau(\boldsymbol{p}, \boldsymbol{W}), \boldsymbol{\Theta})\right] = \mathbb{E}_{p(\boldsymbol{p}, \tilde{\boldsymbol{W}}, \boldsymbol{\Theta})}\left[\frac{\mathbb{E}_{p(\boldsymbol{W}|\tilde{\boldsymbol{W}})}\left[f(\boldsymbol{G}, \boldsymbol{\Theta})p(\boldsymbol{D}, \boldsymbol{\Theta}|\boldsymbol{p}, \boldsymbol{W})\right]}{\mathbb{E}_{p(\boldsymbol{W}|\tilde{\boldsymbol{W}})}\left[p(\boldsymbol{D}, \boldsymbol{\Theta}|\boldsymbol{p}, \boldsymbol{W})\right]}\right] \qquad (17)$$

*where $\boldsymbol{f}$ is the target quantity.*

This proof is in Appendix B.3.

With this proposition, instead of sampling $\boldsymbol{W}$, use SG-MCMC to draw $\tilde{\boldsymbol{W}}$ samples. Similar to Section 4.2, to use SG-MCMC for $\boldsymbol{p}, \tilde{\boldsymbol{W}}, \boldsymbol{\Theta}$, we need their gradient information. The following proposition specifies the required gradients.

**Proposition A.2.** *With the generative model defined as Figure 5, we have*

$$\nabla_{\boldsymbol{p}, \boldsymbol{\Theta}, \tilde{\boldsymbol{W}}} U(\boldsymbol{p}, \tilde{\boldsymbol{W}}, \boldsymbol{\Theta}) = -\nabla_{\boldsymbol{p}} \log p(\boldsymbol{p}) - \nabla_{\boldsymbol{\Theta}} \log p(\boldsymbol{\Theta})$$
$$-\nabla_{\tilde{\boldsymbol{W}}} \log p(\tilde{\boldsymbol{W}}) - \nabla_{\boldsymbol{p}, \boldsymbol{\Theta}, \tilde{\boldsymbol{W}}} \log \mathbb{E}_{p(\boldsymbol{W}|\tilde{\boldsymbol{W}})}[p(\boldsymbol{D}|\boldsymbol{W}, \boldsymbol{p}, \boldsymbol{\Theta})] \qquad (18)$$

The proof is in Appendix B.4.

With these gradients, we can directly plug in existing SG-MCMC samplers to draw samples for $\boldsymbol{p}, \tilde{\boldsymbol{W}}$, and $\boldsymbol{\Theta}$ in joint inference (Algorithm 2).

## B Theory

### B.1 Proof of Theorem 3.1

For completeness, we recite the theorem here.

*Theorem 3.1* (Equivalence of inference in $(\boldsymbol{W}, \boldsymbol{p})$ and binary DAG space). Assume graph $\boldsymbol{G}$ is a binary adjacency matrix representing a DAG and node potential $\boldsymbol{p}$ does not contain the same values, i.e. $p_i \neq p_j \ \forall i, j$. Then, with the induced joint observational distribution $p(\boldsymbol{D}, \boldsymbol{G})$, dataset $\boldsymbol{D}$ and a corresponding prior $p(\boldsymbol{G})$, we have

$$p(\boldsymbol{G}|\boldsymbol{D}) = \int p_\tau(\boldsymbol{p}, \boldsymbol{W}|\boldsymbol{D}) \, \mathbb{1}(\boldsymbol{G} = \tau(\boldsymbol{W}, \boldsymbol{p})) d\boldsymbol{W} d\boldsymbol{p} \qquad (19)$$

if $p(\boldsymbol{G}) = \int p_\tau(\boldsymbol{p}, \boldsymbol{W}) \, \mathbb{1}(\boldsymbol{G} = \tau(\boldsymbol{W}, \boldsymbol{p})) d\boldsymbol{W} d\boldsymbol{p}$, where $p_\tau(\boldsymbol{W}, \boldsymbol{p})$ is the prior, $\mathbb{1}(\cdot)$ is the indicator function and $p_\tau(\boldsymbol{p}, \boldsymbol{W}|D)$ is the posterior distribution over $\boldsymbol{p}, \boldsymbol{W}$.

**Algorithm 2** Joint inference

---

**Input:** dataset $\boldsymbol{D}$, prior $p(\boldsymbol{p}, \tilde{\boldsymbol{W}}, \boldsymbol{\Theta})$, SG-MCMC sampler update $\mathrm{Sampler}(\cdot)$; sampler hyperparameter $\Psi$; training steps $T$.
**Output:** posterior samples $\{\boldsymbol{p}, \tilde{\boldsymbol{W}}, \boldsymbol{\Theta}\}$
Initialize $\boldsymbol{p}_0, \tilde{\boldsymbol{W}}_0, \boldsymbol{\Theta}_0$
**for** $t = 1, \ldots, T$ **do**
    Evaluate gradient $\nabla_{\boldsymbol{p}_{t-1}, \tilde{\boldsymbol{w}}_{t-1}, \boldsymbol{\Theta}_{t-1}} U$ based on Equation (18).
    Update samples $\boldsymbol{p}_t, \tilde{\boldsymbol{W}}_t, \boldsymbol{\Theta}_t = \mathrm{Sampler}(\nabla_{\boldsymbol{p}_{t-1}, \tilde{\boldsymbol{w}}_{t-1}, \boldsymbol{\Theta}_{t-1}} U; \Psi)$
    **if** storing condition met **then**
        $\{\boldsymbol{p}, \tilde{\boldsymbol{W}}, \boldsymbol{\Theta}\} \leftarrow \boldsymbol{p}_t, \tilde{\boldsymbol{W}}_t, \boldsymbol{\Theta}_t$
    **end if**
**end for**

---

To prove this theorem, we first prove the following lemma stating the equivalence of $\tau$ (Equation (5)) to binary DAG space.

**Lemma B.1** (Equivalence of $\tau$ to DAG space). *Consider $d$ random variables, a node potential vector $\boldsymbol{p} \in \mathbb{R}^d$ and a binary matrix $\boldsymbol{W} \in \{0,1\}^{d \times d}$. Then the following holds:*

    *(a) For any $\boldsymbol{W} \in \{0,1\}^{d \times d}$, $\boldsymbol{p} \in \mathbb{R}^d$, $\boldsymbol{G} = \tau(\boldsymbol{W}, \boldsymbol{p})$ is a DAG.*

    *(b) For any DAG $\boldsymbol{G} \in \mathbb{D}$, where $\mathbb{D}$ is the space of all DAGs, there exists a corresponding $(\boldsymbol{W}, \boldsymbol{p})$ such that $\tau(\boldsymbol{W}, \boldsymbol{p}) = \boldsymbol{G}$.*

*Proof.* The main proof directly follows the theorem 2.1 in [72]. For (a), we show the output from $\tau(\boldsymbol{W}, \boldsymbol{p})$ must be a DAG. By leveraging the Lemma 3.4 in [72], we can easily obtain that $\mathrm{Step}(\mathrm{grad}\,\boldsymbol{p})$ emits a binary adjacency matrix representing a DAG. The only difference is that we replace the $\mathrm{ReLU}(\cdot)$ with $\mathrm{Step}(\cdot)$ but the conclusion can be directly generalized.

For (b), we show that for any DAG $\boldsymbol{G}$, there exists a $(\boldsymbol{W}, \boldsymbol{p})$ pair s.t. $\tau(\boldsymbol{W}, \boldsymbol{p}) = \boldsymbol{G}$. To see this, we can observe that $\boldsymbol{p}$ implicitly defines a topological order in the mapping $\tau$. For any $p_i > p_j$, we have $j \to i$ after the mapping $\mathrm{Step}(\mathrm{grad}\,\boldsymbol{p})$. Thus, by leveraging Theorem 3.7 in [72], we obtain that there exists a potential vector $\boldsymbol{p} \in \mathbb{R}^d$ for any DAG $\boldsymbol{G}$ such that

$$(\mathrm{grad}\,\boldsymbol{p})(i,j) > 0 \quad \text{when } G_{ij} = 1$$

Thus, we can choose $\boldsymbol{W}$ in the following way:

$$\boldsymbol{W} = \left\{ \begin{array}{ll} W_{ij} = 0 & \text{if } G_{ij} = 0 \\ W_{ij} = 1 & \text{if } G_{ij} = 1 \end{array} \right.$$

$\square$

Next, let's prove the Theorem 3.1.

*Proof of Theorem 3.1.* From Lemma B.1, we see that the mapping is complete. Namely, the $(\boldsymbol{W}, \boldsymbol{p})$ space can represent the entire DAG space. Next, we show that performing Bayesian inference in $(\boldsymbol{W}, \boldsymbol{p})$ space can also correspond to the inference in DAG space.

Assume we have the prior $p_\tau(\boldsymbol{W}, \boldsymbol{p})$. Then through mapping $\tau$, we implicitly define a prior over the DAG $\boldsymbol{G}$ in the following:

$$p_\tau(\boldsymbol{G}) = \int p_\tau(\boldsymbol{W}, \boldsymbol{p})\, \mathbb{1}(\boldsymbol{G} = \tau(\boldsymbol{W}, \boldsymbol{p})) d\boldsymbol{W}\, d\boldsymbol{p} \tag{20}$$

This basically states that the corresponding prior over $\boldsymbol{G}$ is an accumulation of the corresponding probability associated with $(\boldsymbol{W}, \boldsymbol{p})$ pairs.

Similarly, we can define a corresponding posterior $p_\tau(\boldsymbol{G}|\boldsymbol{D})$:

$$p_\tau(\boldsymbol{G}|\boldsymbol{D}) = \int p_\tau(\boldsymbol{W}, \boldsymbol{p}|\boldsymbol{D})\, \mathbb{1}(\boldsymbol{G} = \tau(\boldsymbol{W}, \boldsymbol{p})) d\boldsymbol{W}\, d\boldsymbol{p} \tag{21}$$

Now, let's show that this posterior $p_\tau(\boldsymbol{G}|\boldsymbol{D}) = p(\boldsymbol{G}|\boldsymbol{D})$ if prior matches, i.e. $p(\boldsymbol{G}) = p_\tau(\boldsymbol{G})$. From Bayes's rule, we can easily write down

$$p_\tau(\boldsymbol{W},\boldsymbol{p}|\boldsymbol{D}) = \frac{p(\boldsymbol{D}|\boldsymbol{G} = \tau(\boldsymbol{W},\boldsymbol{p}))p(\boldsymbol{p},\boldsymbol{W})}{\sum_{\boldsymbol{G}'\in\mathbb{D}}p(\boldsymbol{D},\boldsymbol{G}')} \tag{22}$$

Then, by substituting Equation (22) into Equation (21), we have

$$p_\tau(\boldsymbol{G}|\boldsymbol{D}) = \int \frac{p(\boldsymbol{D}|\boldsymbol{G})p_\tau(\boldsymbol{W},\boldsymbol{p})}{\sum_{\boldsymbol{G}'\in\mathbb{D}}p(\boldsymbol{D},\boldsymbol{G}')} \mathbb{1}(\boldsymbol{G} = \tau(\boldsymbol{W},\boldsymbol{p}))d\boldsymbol{W}d\boldsymbol{p}$$

$$= \frac{\int p(\boldsymbol{D}|\boldsymbol{G})p_\tau(\boldsymbol{W},\boldsymbol{p})\,\mathbb{1}(\boldsymbol{G} = \tau)d\boldsymbol{W}d\boldsymbol{p}}{\sum_{\boldsymbol{G}'\in\mathbb{D}}p(\boldsymbol{D},\boldsymbol{G}')} \tag{23}$$

$$= \frac{p(\boldsymbol{D}|\boldsymbol{G})\int p_\tau(\boldsymbol{W},\boldsymbol{p})\,\mathbb{1}(\boldsymbol{G} = \tau)d\boldsymbol{W}d\boldsymbol{p}}{\sum_{\boldsymbol{G}'\in\mathbb{D}}p(\boldsymbol{D},\boldsymbol{G}')} \tag{24}$$

$$= \frac{p(\boldsymbol{D}|\boldsymbol{G})p_\tau(\boldsymbol{G})}{\sum_{\boldsymbol{G}'\in\mathbb{D}}p(\boldsymbol{D}|\boldsymbol{G}')p_\tau(\boldsymbol{G}')}$$

$$= p(\boldsymbol{G}|\boldsymbol{D}) \tag{25}$$

where Equation (23) is from the fact that $\sum_{\boldsymbol{G}'\in\mathbb{D}}p(\boldsymbol{D},\boldsymbol{G}')$ is independent of $(\boldsymbol{W},\boldsymbol{p})$ due to marginalization. Equation (24) is obtained because $p(\boldsymbol{D}|\boldsymbol{G})$ is also independent of $(\boldsymbol{W},\boldsymbol{p})$ due to (1) $\mathbb{1}(\boldsymbol{G} = \tau(\boldsymbol{W},\boldsymbol{p}))$ and (2) $p(\boldsymbol{D}|\boldsymbol{G})$ is a constant when fixing $\boldsymbol{G}$. Equation (25) is obtained by applying Bayes's rule and $p_\tau(\boldsymbol{G}) = p(\boldsymbol{G})$. □

## B.2 Proof of Theorem 3.2

*Theorem 3.2* (Equivalence of NoCurl formulation). Assuming the conditions in Theorem 3.1 are satisfied. Then, for a given $(\boldsymbol{W},\boldsymbol{p})$, we have

$$\boldsymbol{G} = \boldsymbol{W} \odot \operatorname{Step}(\operatorname{grad}\boldsymbol{p}) = \boldsymbol{W} \odot \left[\boldsymbol{\sigma}^*(\boldsymbol{p})\boldsymbol{L}\boldsymbol{\sigma}^*(\boldsymbol{p})^T\right]$$

where $\boldsymbol{G}$ is a DAG and $\boldsymbol{\sigma}^*(\boldsymbol{p})$ is defined in Equation (7).

To prove this theorem, we need to first prove the following lemma.

**Lemma B.2.** *For any permutation matrix $\boldsymbol{M} \in \Sigma_d$, we have*

$$\operatorname{grad}(\boldsymbol{M}\boldsymbol{p}) = \boldsymbol{M}^T \operatorname{grad}(p)\boldsymbol{M}$$

*where* $\operatorname{grad}$ *is the operator defined in Equation* (3).

*Proof.* By definition of $\operatorname{grad}(\cdot)$, we have

$$\operatorname{grad}(\boldsymbol{M}\boldsymbol{p}) = (\boldsymbol{M}\boldsymbol{p})_i - (\boldsymbol{M}\boldsymbol{p})_j$$

$$= \mathbf{1}(i)^T \boldsymbol{M}\boldsymbol{p} - \mathbf{1}(j)^T \boldsymbol{M}\boldsymbol{p}$$

$$= \boldsymbol{M}_{i,:}\boldsymbol{p} - \boldsymbol{M}_{j,:}\boldsymbol{p}$$

where $\mathbf{1}(i)$ is a one-hot vector with $i^{\text{th}}$ entry 1, and $\boldsymbol{M}_{i,:}$ is the $i^{\text{th}}$ row of matrix $\boldsymbol{M}$. The above is equivalent to computing the $\operatorname{grad}$ with new labels obtained by permuting $\boldsymbol{p}$ with $\boldsymbol{M}$. Therefore, we can see that $\operatorname{grad}(\boldsymbol{M}\boldsymbol{p})$ can be computed by permuting the original $\operatorname{grad}(\boldsymbol{p})$ by matrix $\boldsymbol{M}$.

$$\operatorname{grad}(\boldsymbol{M}\boldsymbol{p}) = \boldsymbol{M}^T \operatorname{grad}(\boldsymbol{p})\boldsymbol{M}$$

□

*Proof of Theorem 3.2.* Since $\boldsymbol{W}$ plays the same role in both formulations, we focus on the equivalence of $\operatorname{Step}(\operatorname{grad}(\cdot))$.

Define a sorted $\tilde{\boldsymbol{p}} = \boldsymbol{\sigma}\boldsymbol{p}$, where $\boldsymbol{\sigma} \in \Sigma_d$, such that for $i < j$, we have $\tilde{p}_i > \tilde{p}_j$. Namely, $\boldsymbol{\sigma}$ is a permutation matrix. Thus, we have

$$\operatorname{grad}(\boldsymbol{p}) = \operatorname{grad}(\boldsymbol{\sigma}^T\tilde{\boldsymbol{p}}).$$

By Lemma B.2, we have
$$\mathrm{grad}(\boldsymbol{\sigma}^T\tilde{\boldsymbol{p}}) = \boldsymbol{\sigma}\,\mathrm{grad}(\tilde{p})\boldsymbol{\sigma}^T.$$
Since $\tilde{\boldsymbol{p}}$ is an ordered vector. Therefore, $\mathrm{grad}(\tilde{\boldsymbol{p}})$ is a skew-symmetric matrix with a positive lower half part.

Therefore, we have
$$\mathrm{Step}(\mathrm{grad}(\boldsymbol{p})) = \mathrm{Step}(\boldsymbol{\sigma}\,\mathrm{grad}(\tilde{p})\boldsymbol{\sigma}^T) = \boldsymbol{\sigma}\,\mathrm{Step}(\mathrm{grad}(\tilde{p}))\boldsymbol{\sigma}^T = \boldsymbol{\sigma}\boldsymbol{L}\boldsymbol{\sigma}^T$$
This is true because $\boldsymbol{\sigma}$ is just a permutation matrix that does not alter the sign of $\mathrm{grad}(\tilde{\boldsymbol{p}})$.

Since $\boldsymbol{\sigma}$ is a permutation matrix that sort $\boldsymbol{p}$ value in a ascending order, from Lemma 1 in [8], we have
$$\boldsymbol{\sigma} = \arg\max_{\boldsymbol{\sigma}'\in\boldsymbol{\Sigma}_d}\boldsymbol{p}^T(\boldsymbol{\sigma}'\boldsymbol{o})$$

$\square$

## B.3  Proof of Proposition A.1

*Proof.*
$$\mathbb{E}_{p(\boldsymbol{p},\boldsymbol{W},\boldsymbol{\Theta}|\boldsymbol{D})}\left[f(\boldsymbol{G}=\tau(\boldsymbol{p},\boldsymbol{W}),\boldsymbol{\Theta})\right]$$
$$= \int p(\boldsymbol{p},\boldsymbol{W},\boldsymbol{\Theta},\tilde{\boldsymbol{W}}|\boldsymbol{D})f(\boldsymbol{G},\boldsymbol{\Theta})d\boldsymbol{p}d\boldsymbol{W}d\boldsymbol{\Theta}d\tilde{\boldsymbol{W}}$$
$$= \int p(\boldsymbol{p},\tilde{\boldsymbol{W}},\boldsymbol{\Theta}|\boldsymbol{D})p(\boldsymbol{W}|\boldsymbol{p},\boldsymbol{\Theta},\tilde{\boldsymbol{W}},\boldsymbol{D})f(\boldsymbol{G},\boldsymbol{\Theta})d\boldsymbol{p}d\boldsymbol{W}d\boldsymbol{\Theta}d\tilde{\boldsymbol{W}}$$
$$= \mathbb{E}_{p(\boldsymbol{p},\tilde{\boldsymbol{W}},\boldsymbol{\Theta}|\boldsymbol{D})}\left[\frac{\int p(\boldsymbol{D}|\boldsymbol{p},\boldsymbol{\Theta},\boldsymbol{W})p(\boldsymbol{p})p(\tilde{\boldsymbol{W}})p(\boldsymbol{W}|\tilde{\boldsymbol{W}})p(\boldsymbol{\Theta}|\boldsymbol{p},\boldsymbol{W})f(\boldsymbol{G},\boldsymbol{\Theta})d\boldsymbol{W}}{\int p(\boldsymbol{D}|\boldsymbol{p},\boldsymbol{\Theta},\boldsymbol{W})p(\boldsymbol{p})p(\tilde{\boldsymbol{W}})p(\boldsymbol{W}|\tilde{\boldsymbol{W}})p(\boldsymbol{\Theta}|\boldsymbol{p},\boldsymbol{W})d\boldsymbol{W}}\right]$$
$$= \mathbb{E}_{p(\boldsymbol{p},\tilde{\boldsymbol{W}},\boldsymbol{\Theta})}\left[\frac{\mathbb{E}_{p(\boldsymbol{W}|\tilde{\boldsymbol{W}})}\left[f(\boldsymbol{G},\boldsymbol{\Theta})p(\boldsymbol{D},\boldsymbol{\Theta}|\boldsymbol{p},\boldsymbol{W})\right]}{\mathbb{E}_{p(\boldsymbol{W}|\tilde{\boldsymbol{W}})}\left[p(\boldsymbol{D},\boldsymbol{\Theta}|\boldsymbol{p},\boldsymbol{W})\right]}\right]$$

$\square$

## B.4  Proof of Proposition A.2

*Proof.*
$$\nabla_{\boldsymbol{p}}U(\boldsymbol{p},\tilde{\boldsymbol{W}},\boldsymbol{\Theta}) = -\nabla_{\boldsymbol{p}}\log p(\boldsymbol{p},\tilde{\boldsymbol{W}},\boldsymbol{\Theta},\boldsymbol{D})$$
$$= -\nabla_{\boldsymbol{p}}\log p(\boldsymbol{p}) - \nabla_{\boldsymbol{p}}\log p(\tilde{\boldsymbol{W}},\boldsymbol{\Theta},\boldsymbol{D}|\boldsymbol{p})$$
$$= -\nabla_{\boldsymbol{p}}\log p(\boldsymbol{p}) - \frac{\nabla_{\boldsymbol{p}}\int p(\boldsymbol{D}|\boldsymbol{W},\boldsymbol{p},\boldsymbol{\Theta})p(\boldsymbol{\Theta}|\boldsymbol{p},\boldsymbol{W})p(\boldsymbol{W}|\tilde{\boldsymbol{W}})p(\tilde{\boldsymbol{W}})d\boldsymbol{W}}{\int p(\boldsymbol{D}|\boldsymbol{W},\boldsymbol{p},\boldsymbol{\Theta})p(\boldsymbol{\Theta}|\boldsymbol{p},\boldsymbol{W})p(\boldsymbol{W}|\tilde{\boldsymbol{W}})p(\tilde{\boldsymbol{W}})d\boldsymbol{W}}$$
$$= -\nabla_{\boldsymbol{p}}\log p(\boldsymbol{p}) - \frac{\nabla_{\boldsymbol{p}}\mathbb{E}_{p(\boldsymbol{W}|\tilde{\boldsymbol{W}})}\left[p(\boldsymbol{D}|\boldsymbol{W},\boldsymbol{p},\boldsymbol{\Theta})\right]}{\mathbb{E}_{p(\boldsymbol{W}|\tilde{\boldsymbol{W}})}\left[p(\boldsymbol{D}|\boldsymbol{W},\boldsymbol{p},\boldsymbol{\Theta})\right]}$$
$$= -\nabla_{\boldsymbol{p}}\log p(\boldsymbol{p}) - \nabla_{\boldsymbol{p}}\log\mathbb{E}_{p(\boldsymbol{W}|\tilde{\boldsymbol{W}})}\left[p(\boldsymbol{D}|\boldsymbol{W},\boldsymbol{p},\boldsymbol{\Theta})\right]$$

Other gradient $\nabla_{\tilde{\boldsymbol{W}}}U$ and $\nabla_{\boldsymbol{\Theta}}U$ can be derived using the similar approach, which concludes the proof. $\square$

## B.5  Proof of Proposition 4.1

*Proof of Proposition 4.1.* By definition, we have easily have
$$\nabla_{\boldsymbol{p}}U = -\nabla_{\boldsymbol{p}}\log p(\boldsymbol{p},\boldsymbol{W},\boldsymbol{\Theta},\boldsymbol{D})$$
$$= -\nabla_{\boldsymbol{p}}\log p(\boldsymbol{p},\boldsymbol{W}) - \nabla_{\boldsymbol{p}}\log p(\boldsymbol{D},\boldsymbol{\Theta}|\tau(\boldsymbol{W},\boldsymbol{p}))$$
$$= -\nabla_{\boldsymbol{p}}\log p(\boldsymbol{p},\boldsymbol{W}) - \nabla_{\boldsymbol{p}}\log p(\boldsymbol{D}|\boldsymbol{\Theta},\tau(\boldsymbol{W},\boldsymbol{p})) + \underbrace{\nabla_{\boldsymbol{p}}\log p(\boldsymbol{\Theta}|\tau(\boldsymbol{p},\boldsymbol{W}))}_{0}$$

Similarly, we have

$$\begin{aligned}
\nabla_{\boldsymbol{\Theta}}U &= -\nabla_{\boldsymbol{\Theta}}\log p(\boldsymbol{p},\boldsymbol{W},\boldsymbol{\Theta},\boldsymbol{D}) \\
&= -\nabla_{\boldsymbol{\Theta}}\log p(\boldsymbol{D}|\boldsymbol{\Theta},\tau(\boldsymbol{W},\boldsymbol{p})) - \nabla_{\boldsymbol{\Theta}}\log p(\boldsymbol{\Theta}|\boldsymbol{p},\boldsymbol{W}) - \underbrace{\nabla_{\boldsymbol{\Theta}}\log p(\boldsymbol{p},\boldsymbol{W})}_{0} \\
&= -\nabla_{\boldsymbol{\Theta}}\log p(\boldsymbol{D}|\boldsymbol{\Theta},\tau(\boldsymbol{W},\boldsymbol{p})) - \nabla_{\boldsymbol{\Theta}}\log p(\boldsymbol{\Theta})
\end{aligned}$$

$\square$

### B.6 Derivation of ELBO

$$\begin{aligned}
\log p(\boldsymbol{p},\boldsymbol{\Theta},\boldsymbol{D}) &= \log \int p(\boldsymbol{p},\boldsymbol{\Theta},\boldsymbol{D},\boldsymbol{W})d\boldsymbol{W} \\
&= \log \int \frac{q_\phi(\boldsymbol{W}|\boldsymbol{p})}{q_\phi(\boldsymbol{W}|\boldsymbol{p})}p(\boldsymbol{p},\boldsymbol{\Theta},\boldsymbol{D},\boldsymbol{W})d\boldsymbol{W} \\
&\geq \int q_\phi(\boldsymbol{W}|\boldsymbol{p})\log p(\boldsymbol{p},\boldsymbol{\Theta},\boldsymbol{D}|\boldsymbol{W})d\boldsymbol{W} + \int q_\phi(\boldsymbol{W}|\boldsymbol{p})\log \frac{p(\boldsymbol{W})}{q_\phi(\boldsymbol{W}|\boldsymbol{p})}d\boldsymbol{W} \quad (26) \\
&= \mathbb{E}_{q_\phi(\boldsymbol{W}|\boldsymbol{p})}\left[\log p(\boldsymbol{p},\boldsymbol{\Theta},\boldsymbol{D}|\boldsymbol{W})\right] - D_{\mathrm{KL}}\left[q_\phi(\boldsymbol{W}|\boldsymbol{p})\|p(\boldsymbol{W})\right]
\end{aligned}$$

where the Equation (26) is obtained by Jensen's inequality.

## C    SG-MCMC Update

Assume we want to draw samples $\boldsymbol{p} \sim p(\boldsymbol{p}|\boldsymbol{D},\boldsymbol{W},\boldsymbol{\Theta}) \propto \exp(-U(\boldsymbol{p},\boldsymbol{W},\boldsymbol{\Theta}))$ with $U(\boldsymbol{p},\boldsymbol{W},\boldsymbol{\Theta}) = -\log p(\boldsymbol{p},\boldsymbol{W},\boldsymbol{\Theta})$, we can compute $U$ by

$$U(\boldsymbol{p},\boldsymbol{W},\boldsymbol{\Theta}) = -\sum_{n=1}^{N}\log p(\boldsymbol{x}_n|\boldsymbol{G}=\tau(\boldsymbol{W},\boldsymbol{p}),\boldsymbol{\Theta}) - \log p(\boldsymbol{p},\boldsymbol{W},\boldsymbol{\Theta}) \quad (27)$$

In practice, we typically use mini-batches $\mathcal{S}$ instead of the entire dataset $\boldsymbol{D}$. Therefore, an approximation is

$$\tilde{U}(\boldsymbol{p},\boldsymbol{W},\boldsymbol{\Theta}) = -\frac{|\boldsymbol{D}|}{|\mathcal{S}|}\sum_{n\in\mathcal{S}}\log p(\boldsymbol{x}_n|\boldsymbol{G}=\tau(\boldsymbol{W},\boldsymbol{p}),\boldsymbol{\Theta}) - \log p(\boldsymbol{p},\boldsymbol{W},\boldsymbol{\Theta}) \quad (28)$$

where $|\mathcal{S}|$ and $|\boldsymbol{D}|$ are the minibatch and dataset sizes, respectively.

[28] uses the preconditioning technique on *stochastic gradient Hamiltonian Monte Carlo*(SG-HMC), similar to the preconditioning technique in [41]. In particular, they use a moving-average approximation of diagonal Fisher information to adjust the momentum. The transition dynamics at step $t$ with EM discretization is

$$\begin{aligned}
B &= \frac{1}{2}l \\
\boldsymbol{V}_t &= \beta_2\boldsymbol{V}_{t-1} + (1-\beta_2)\nabla_{\boldsymbol{p}}\tilde{U}(\boldsymbol{p},\boldsymbol{W},\boldsymbol{\Theta}) \odot \nabla_{\boldsymbol{p}}\tilde{U}(\boldsymbol{p},\boldsymbol{W},\boldsymbol{\Theta}) \\
g_t &= \frac{1}{\sqrt{1+\sqrt{\boldsymbol{V}_t}}} \\
\boldsymbol{r}_t &= \beta_1\boldsymbol{r}_{t-1} - lg_t\nabla_{\boldsymbol{p}}\tilde{U}(\boldsymbol{p},\boldsymbol{W},\boldsymbol{\Theta}) + l\frac{\partial g_t}{\partial \boldsymbol{p}_t} + s\sqrt{2l(\frac{1-\beta_1}{l}-B)\eta} \\
\boldsymbol{p}_t &= \boldsymbol{p}_{t-1} + lg_t\boldsymbol{r}_t
\end{aligned} \quad (29)$$

where $l^2$ is the learning rate; $(\beta_1,\beta_2)$ controls the preconditioning decay rate, $\eta$ is the Gaussian noise with 0 mean and unit variance, and $s$ is the hyperparameter controlling the level of injected noise to SG-MCMC. Throughout the paper, we use $(\beta_1,\beta_2) = (0.9,0.99)$ for all experiments.

# D Experimental Settings

## D.1 Baselines

For all the experimental settings, we compare with the following baselines:

- Bootstrap GES (**BGES**) [20, 14] is a bootstrap based quasi-Bayesian approach for linear Gaussian models which first resamples with replacement data points at random and then estimates a linear SCM using the GES algorithm [14] for each bootstrap set. GES is a score based approach to learn a point estimate of a linear Gaussian SCM. For all the experimental settings, we use 50 bootstrap sets.

- Differentiable DAG Sampling (**DDS**) is a VI based approach to learn distribution over DAGs and a point estimate over the nonlinear functional parameters. DDS performs inference on the node permutation matrices, thus directly generating DAGs. Gumbel-sinkhorn [47] is used for obtaining valid gradients and Hungarian algorithm is used for the straight-through gradient estimator [7]. In the author provided implementation, for evaluation, a single permutation matrix is sampled and the logits of the edge beliefs are directly thresholded. In this work, in order to make the comaprison fair to Bayesian learning methods, we directly sample the binary adjacency matrix based on the edge logits.

- **BCD** Nets [16] is a VI based fully Bayesian structure learning approach for linear causal models. BCD performs inference on both the node permutations through the Gumbel-sinkhorn [47] operator as well as the model parameters through a VI distribution. Both DDS and BCD nets operate directly on full rank initializations to the Gumbel-sinkhorn operator, unlike our rank-1 initialization, which saves computations in practice.

- **DIBS** [43] uses SVGD [42] with the DAG regularizer [77] and bilinear embeddings to perform inference over both linear and nonlinear causal models. As our data generation process involves SCM with unequal noise variance, we extend DIBS framework with an inference over noise variance using SVGD, similar to the original paper.

While DIBS and DDS can handle nonlinear parameterization, approaches like BGES and BCD, which are primarily designed for linear models still give competitive results when applied on nonlinear data. Given that there are limited number of baselines in the nonlinear case, and DIBS being the only fully Bayesian nonlinear baseline, we compare with BGES and BCD for all settings despite their model misspecification.

## D.2 Evaluation Metrics

For higher dimensional settings with nonlinear models, the true posterior is intractable. While in general it is hard to evaluate the posterior inference quality in high dimensions, prior work has suggested to evaluate on proxy metrics which we adopt in this work as well [43, 22, 4]. In particular, we evaluate the following metrics:

- $\mathbb{E}$**-SHD**: Structural Hamming Distance (SHD) measures the hamming distance between graphs. In particular, it is a measure of number of edges that are to be added, removed or reversed to get the ground truth from the estimated graph. Since we have a posterior distribution $q(\boldsymbol{G})$ over graphs, we measure the *expected* SHD:

$$\mathbb{E}\text{-SHD} := \mathbb{E}_{\boldsymbol{G} \sim q(\boldsymbol{G})}[\text{SHD}(\boldsymbol{G}, \boldsymbol{G}^{GT})] \approx \frac{1}{N_e} \sum_{i=1}^{N_e} [\text{SHD}(\boldsymbol{G}^{(i)}, \boldsymbol{G}^{GT})] \quad , \text{with } \boldsymbol{G}^{(i)} \sim q(\boldsymbol{G})$$

  where $\boldsymbol{G}^{GT}$ is the ground-truth causal graph.

- **Edge F1**: It is F1 score of each edge being present or absent in comparison to the true edge set, averaged over all edges.

- **NLL**: We also measure the negative log-likelihood of the held-out data, which is also typically used as a proxy metric on evaluating the posterior inference quality [26, 44, 63].

The first two metrics measure the goodness of the graph posterior while the NLL measures the goodness of the joint posterior over the entire causal model.

For $d = 5$ with linear models (unequal noise variance, identifiable upto MEC [53, 34]), we evaluate the following metrics:

- **MMD True Posterior:** Since the true posterior is tractable, we compare the approximation with the ground truth using a Maximum Mean Discrepancy (MMD) metric [31]. If $\mathrm{P} := p(\boldsymbol{G} \mid \boldsymbol{D})$ is the marginalized true posterior over graphs and Q is the approximated posterior over graphs, then the MMD between these two distributions is defined as:

$$\mathrm{MMD}^2(\mathrm{P}, \mathrm{Q}) = \mathbb{E}_{\boldsymbol{G} \sim \mathrm{P}}[k(\boldsymbol{G}, \boldsymbol{G})] + \mathbb{E}_{\boldsymbol{G} \sim \mathrm{Q}}[k(\boldsymbol{G}, \boldsymbol{G})] - 2\mathbb{E}_{\boldsymbol{G} \sim \mathrm{P}, \boldsymbol{G}' \sim \mathrm{Q}}[k(\boldsymbol{G}, \boldsymbol{G}')]$$

  where $k(\boldsymbol{G}, \boldsymbol{G}') = 1 - \frac{H(\boldsymbol{G}, \boldsymbol{G}')}{d^2}$ is the Hamming kernel, and $H$ is the Hamming distance between $\boldsymbol{G}$ and $\boldsymbol{G}'$. This requires just the samples from the true posterior and the model. For calculating the true posterior which involves marginalization of the model parameters, appropriate prior over these parameters are required. This is ensured by using BGe score [23, 39] which places a Gaussian Wishart prior on the parameters. This leads to closed form marginal likelihood which is distribution equivalent, i.e. all graphs within the MEC will have the same likelihood. In addition, due to the low dimensionality ($d = 5$), we can enumerate all possible DAGs and compute the normalizing constant $p(\boldsymbol{D})$. We refer to [23] for details. This metric has been used in prior work [3].

- **$\mathbb{E}$-CPDAG SHD:** An MEC can be represented by a Completed Partially Directed Acyclic Graph (CPDAG) [54] which contains both directed edges and arcs (undirected edges). When causal relations between certain set of variables can be established, a directed edge is present. If there is an association between a certain set of variables for which causal direction is not identifiable, an undirected edge is present. For any graph, it has a corresponding CPDAG associated to the MEC which it belongs to. Since the ground truth graph is identifiable only upto MEC, we compare the (structural) Hamming distance between the graph posterior and the CPDAG of the ground truth. This is done by computing the $\mathbb{E}$-CPDAG SHD:

$$\mathbb{E}\text{-CPDAG SHD} := \mathbb{E}_{\boldsymbol{G} \sim q(\boldsymbol{G})}[\mathrm{SHD}(\boldsymbol{G}_{\mathrm{CPDAG}}, \boldsymbol{G}_{\mathrm{CPDAG}}^{GT})] \approx \frac{1}{N_e} \sum_{i=1}^{N_e} [\mathrm{SHD}(\boldsymbol{G}_{\mathrm{CPDAG}}^{(i)}, \boldsymbol{G}_{\mathrm{CPDAG}}^{GT})]$$

  with $\boldsymbol{G}^{(i)} \sim q(\boldsymbol{G})$ and $\boldsymbol{G}_{CPDAG}^{GT}$ is the ground-truth CPDAG.

### D.3 Synthetic Data

As knowledge of ground truth graph is not possible in many real world settings, it is standard across causal discovery to benchmark in synthetic data settings. Following prior work, we generate synthetic data by first sampling a DAG at random from either Erdos-Rènyi (ER) [19] or Scale-Free (SF) [5] family. For $d = 5$, we ensure that the graphs have $d$ edges in expectation and $2d$ edges for $d > 5$. The ground truth parameters for linear functions are drawn at random from a fixed range of $[0.5, 1.5]$. For nonlinear models, the nonlinear functions are defined by randomly initialized Multi-Layer Perceptrons (MLP) with a single hidden layer of 5 nodes and ReLU nonlinearity. The variance of the exogenous Gaussian noise variable is drawn from an Inverse Gamma prior with concentration $\alpha = 1.5$ and rate $\beta = 1$. For $d = 5$ linear case, we sample at random $N = 500$ samples from the SCM for training and $N = 100$ for held-out evaluation. For higher dimensional settings, we consider $N = 5000$ random samples for training and $N = 1000$ samples for held-out evaluation. For all settings, we evaluate on 30 random datasets.

### D.4 Hyperparameter Selection

In this section, we will give the details our how to select the hyperparameters for our method and all the baseline models.

We employ a cross-validation-like procedure for hyperparameter tuning in BayesDAG and DIBS to optimize MMD true posterior (for $d = 5$ linear setting) and $\mathbb{E}-$SHD value (for nonlinear setting). For each ER and SF dataset with varying dimensions, we initially generate five tuning datasets. After determining the optimal hyperparameters, we fix them and evaluate the models on 30 test datasets. For DDS, we adopt the hyperparameters provided in the original paper [11]. BCD and BGES do not necessitate hyperparameter tuning since BCD already incorporates the correct prior graph for ER and SF datasets. For semi-synthetic Syntren and real world Sachs protein cells datasets, we assume the

| BayesDAG | | | | |
|---|---|---|---|---|
| | $\lambda_s$ | Scale $p$ | Scale $\Theta$ | Sparse Init. |
| linear ER $d = 5$ | 50 | 0.001 | 0.001 | False |
| linear SF $d = 5$ | 50 | 0.01 | 0.001 | False |
| nonlinear ER $d = 20$ | 300 | 0.01 | 0.01 | False |
| nonlinear SF $d = 20$ | 200 | 0.1 | 0.1 | False |
| nonlinear ER $d = 30$ | 500 | 1 | 0.01 | False |
| nonlinear SF $d = 30$ | 300 | 0.01 | 0.01 | False |
| nonlinear ER $d = 50$ | 500 | 0.01 | 0.01 | True |
| nonlinear SF $d = 50$ | 300 | 0.1 | 0.01 | False |
| nonlinear ER $d = 70$ | 700 | 0.1 | 0.01 | True |
| nonlinear SF $d = 70$ | 300 | 0.01 | 0.01 | False |
| nonlinear ER $d = 100$ | 700 | 0.1 | 0.01 | False |
| nonlinear SF $d = 100$ | 700 | 0.1 | 0.01 | False |
| SynTren | 300 | 0.1 | 0.01 | False |
| Sachs Protein Cells | 1200 | 0.1 | 0.01 | False |

Table 3: The hyperparameter selection for `BayesDAG` for each setting.

| DIBS | | | | |
|---|---|---|---|---|
| | $\alpha$ | $h$ latent | $h_\theta$ | $h_\sigma$ |
| linear ER $d = 5$ | 0.02 | 5 | 1000 | 1 |
| linear SF $d = 5$ | 0.02 | 15 | 500 | 1 |
| nonlinear ER $d = 20$ | 0.02 | 5 | 1500 | 10 |
| nonlinear SF $d = 20$ | 0.2 | 5 | 1500 | 10 |
| nonlinear ER $d = 30$ | 0.2 | 5 | 500 | 1 |
| nonlinear SF $d = 30$ | 0.2 | 5 | 1000 | 1 |
| nonlinear ER $d = 50$ | 0.2 | 5 | 500 | 10 |
| nonlinear SF $d = 50$ | 0.2 | 5 | 1500 | 1 |
| SynTren | 0.2 | 5 | 500 | 10 |
| Sachs Protein Cells | 0.2 | 5 | 500 | 10 |

Table 4: The hyperparameter selection for DIBS for each setting.

number of edges in the ground truth graphs are known and we tune our hyperparameters to produce roughly correct number of edges. BCD and DIBS also assume access to the ground truth edge number and use the graph prior to enforce the number of edges.

**Network structure** We use one hidden layer MLP with hidden size of $\max(4 * d, 64)$ for the nonlinear functional relations, where $d$ is the dimensionalilty of dataset. We use **LeakyReLU** as the activation function. We also enable the **LayerNorm** and **residual connections** in the network. In particular, for variational network $\mu_\phi$ in `BayesDAG`, we apply the **LayerNorm** on $p$ before inputting it to the network. We use 2 hidden layer MLP with size 48, **LayerNorm** and **residual connections** for $\mu_\phi$.

**Sparse initialization for** `BayesDAG` For `BayesDAG`, we additionally allow sparse initialization by sampling a sparse $W$ from the $\mu_\phi$. This can be achieved by substracting a constant 1 from the existing logits (i.e. the output from $\mu_\phi$).

**Other hyperparameters** For `BayesDAG`, we run 10 parallel SG-MCMC chains for $p$ and $\Theta$. We implement an adaptive sinkhorn iteration where the iteration automatically stops when the sum of rows and columns are closed to 1 within the threshold 0.001 (upto a maximum of 3000 iterations). Typically, we found this to require only around 300 iterations. We set the sinkhorn temperature $t$ to be 0.2. For the reparametrization of $W$ matrix with Gumbel-softmax trick, we use temperature 0.2. During evaluation, we use 100 SG-MCMC particles extracted from the particle buffer. We use 0.0003 for SG-MCMC learning rate $l$ and batch size 512. We run 700 epochs to make sure the model is fully converged.

Table 5: Walltime results (in minutes, rounded to the nearest minute) of the runtime of different approaches on a single 40GB A100 NVIDIA GPU. The N/A fields indicate that the corresponding method cannot be run within the memory constraints of a single GPU.

|  | d=30 | d=50 | d=70 | d=100 |
|---|---|---|---|---|
| BaDAG (**Ours**)(Bayesian, Nonlinear) | 171 | 238 | 261 | 448 |
| DIBS (Bayesian, Nonlinear) | 187 | 350 | N/A | N/A |
| BGES (Quasi-Bayesian, Linear) | 2 | 3 | 6 | 11 |
| BCD (Bayesian, Linear) | 252 | 328 | 418 | 600 |
| DDS (Quasi-Bayesian, Nonlinear) | 92 | 130 | 174 | N/A |

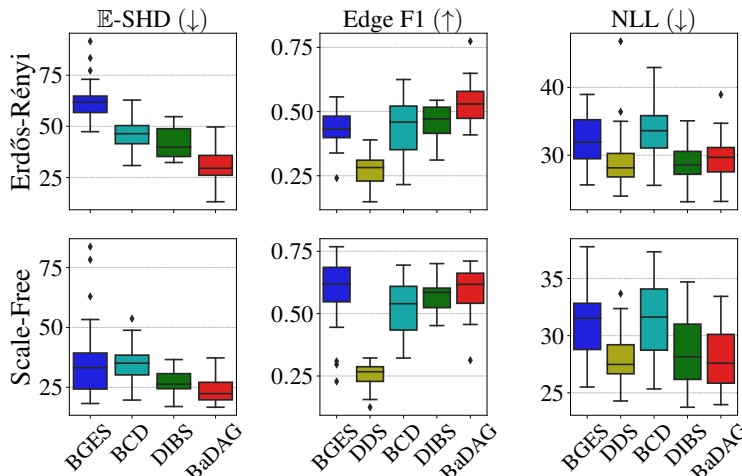

Figure 6: Posterior inference of both graph and functional parameters on synthetic datasets of nonlinear causal models with $d = 20$ variables. `BayesDAG` gives best results across all metrics. ↓ denotes lower is better and ↑ denotes higher is better. For the sake of clarity, DDS has been omitted for $\mathbb{E}$-SHD due to its significantly inferior performance on this metric.

For DIBS, we can only use 20 SVGD particles for evaluation due to the quadratic scaling with the number of particles. We use 0.1 for Gumbel-softmax temperature. We run 10000 epochs for convergence. The learning rate is selected as 0.01.

Table 3 shows the hyperparameter selection for `BayesDAG`. Table 4 shows the hyperparameter selection for DIBS.

# E    Additional Results

## E.1    Walltime Comparison

Table 5 presents walltime comparison of different methods. Our method converges faster while being scalable w.r.t. DIBS, the nonlinear Bayesian causal discovery baseline. Other methods like BGES and DDS, while faster, perform much worse in terms of uncertainty quantification. In addition BGES is limited to linear model and DDS is not a fully Bayesian method.

## E.2    Performance with higher dimensional datasets

Full results for all the metrics for settings $d = 20$, $d = 70$ and $d = 100$ for nonlinear settings are presented in Figure 6, Figure 7 and Figure 8.

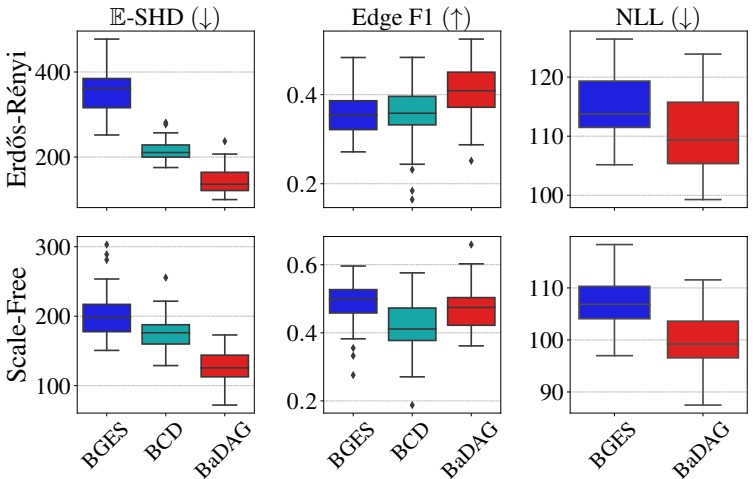

Figure 7: Posterior inference of both graph and functional parameters on synthetic datasets of nonlinear causal models with $d = 70$ variables. `BayesDAG` gives best results across most metrics. ↓ denotes lower is better and ↑ denotes higher is better. As DIBS and DDS are computationally prohibitive to run for this setting, it has been omitted. BCD has been omitted for NLL as we observed that it performs significantly worse.

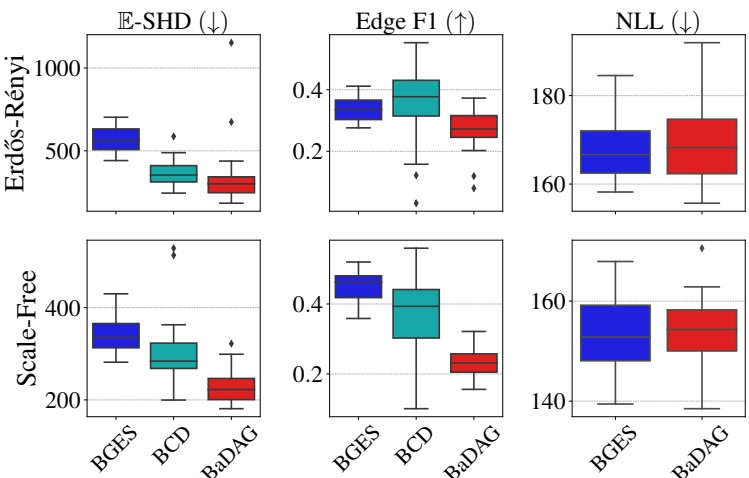

Figure 8: Posterior inference of both graph and functional parameters on synthetic datasets of nonlinear causal models with $d = 100$ variables. `BayesDAG` gives best results across $\mathbb{E}$-SHD, comparable across NLL but slightly worse for Edge F1. ↓ denotes lower is better and ↑ denotes higher is better. As DIBS and DDS are computationally prohibitive to run for this setting, it has been omitted. BCD has been omitted for NLL as we observed that it performs significantly worse.

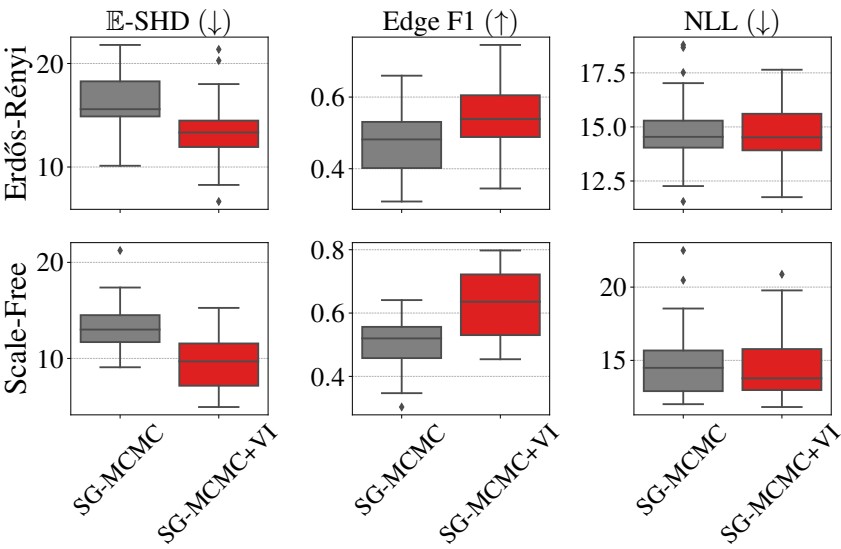

Figure 9: Performance comparison of SG-MCMC+VI v.s. fully SG-MCMC with $\tilde{W}$ for $d = 10$ variables.

### E.3 Performance of SG-MCMC with Continuous Relaxation

We compare the performance of SG-MCMC+VI and SG-MCMC with $\tilde{W}$ on $d = 10$ ER and SF graph settings. Figure 9 shows the performance comparison. We can observe that SG-MCMC+VI generally outperforms its counterpart in most of the metrics. We hypothesize that this is because VI network $\mu_\phi$ couples $p$ and $W$. This coupling effect is crucial since the changes in $p$ results in the change of permutation matrix, where the $W$ can immediately respond to this change through $\mu_\phi$. On the other hand, $\tilde{W}$ can only respond to this change through running SG-MCMC steps on $\tilde{W}$ with fixed $p$. In theory, this is the most flexible approach since this coupling do not requires parametric form like $\mu_\phi$. However in practice, we cannot run many SG-MCMC steps with fixed $p$ for convergence, which results in the inferior performance.

## F   Code and License

For the baselines, we use the code from the following repositories:

- BGES: We use the code from [2] from the repository https://github.com/agrawalraj/active_learning (No license included).
- DDS: We use the code from the official repository https://github.com/sharpenb/Differentiable-DAG-Sampling (No license included).
- BCD: We use the code from the official repository https://github.com/ermongroup/BCD-Nets (No license included).
- DIBS: We use the code from the official repository https://github.com/larslorch/dibs (MIT license).

Additionally for the Syntren [67] and Sachs Protein Cells [59] datasets, we use the data provided with repository https://github.com/kurowasan/GraN-DAG (MIT license).

## G   Broader Impact Statement

This work is concerned with understanding cause and effects from data, which has potential applications in empirical sciences, economics and machine perception. Understanding causal relationships

can improve fairness in decision making, understand biases which might be present in the data and answering causal queries. As such, we envision this line of work to not have any significant negative impact.

