# OpenReview forum: "BayesDAG: Gradient-Based Posterior Inference for Causal Discovery"
_NeurIPS.cc/2023/Conference — NeurIPS 2023 poster_

### Official Review · Reviewer_R7H9 · 2023-06-30

**Soundness:** 3 good
**Presentation:** 3 good
**Contribution:** 3 good
**Rating:** 5
**Confidence:** 3

**Summary:**

BayesDAG proposes a hybrid SG-MCMC sampling and variational inference for drawing samples from the posterior distribution of DAGs in the context of Bayesian structure learning.
This work is closely related to a recently published "Yu et al., Dags with no curl, 2021" [NoCurl] where the space of DAGs is converted to the space of a skew-symmetric matrix, "W", and a potential vector,"p".
The main difference is that [NoCurl] focuses on an optimization setting (to return a most probable DAG) while the focus of the current paper is on Bayesian inference (via sampling).
They also mention that (for reasons that are not entirely clear to me) NoCurl approach optimization is challenging (due to uninformative gradients) and that this is due to the fact that the entries of the skew-symmetric matrix, W are continuous.
To address this problem, they replace the continuous matrix W with a binary matrix.
For this purpose, they slightly modify the theory presented in NoCurl (namely, replacing relu with step function). Then they reformulate the problem as a  matrix permutation setting and finally propose an approximate solution for the latter formulation via Sinkhorn approach to learn latent permutations by the Gumble trick.

**Strengths:**

1. This is a well-written paper addressing an important problem i.e. Bayesian structure learning.
2. The proposed approach is an interesting (albeit sophisticated) combination and modification of several algorithms and recent advances in the field.
3. Even though I did not follow why the continuous matrix W had to be replaced by a binary matrix in the first place, but I found the way they did it and managed to approximate its gradient, very interesting.


**Weaknesses:**

1. As I mentioned in the summary section, a key contribution of this paper is the insight that it is better to replace the continuous matrix W (of NoCurl algorithm) with a binary matrix (see lines 101-104). But their justification for this claim does not seem convincing to me and should be explained better. To be more concrete:
(a) Why should replacing a continuous matrix with a discrete matrix be helpful when we are relying on gradient information for optimization and sampling?
(b) In line 100 they mention a "reported failure" of NoCurl. It would be great if the authors would provide a reference to where this failure is reported (Or if it is reported in the original NoCurl paper, the relevant section).

2. Given the close link of the present paper with the NoCurl approach, it would be great if the authors would compare their algorithm (with binary W) with an alternative approach where just like NoCurl, a continuous W would be used (and relu instead of step function, etc).

3. I see references to repositories that the authors have used but no link to their own code. Given that implementation of the proposed algorithm from scratch is by no means trivial, I encourage the authors to provide the code. Both for facilitating other researchers to use their algorithm as well as allowing the reviewers to check the reproducibility of the reported results.




**Questions:**

Given that the Sinkhorn approach is approximate, how do you guarantee the DAGness of your graphs ?

---

> ### Author Rebuttal · Authors · 2023-08-09
>
> Thanks for your constructive feedback for our paper. We will address your concerns in the following:
>
> 1. **Advantages compared to no-curl**: Sorry about the ambiguity. We will make it more clearer in the revised paper. There are several reasons on why the binary matrix is better than continuous one: (i) Note that $ReLU(grad \pmb{p})$ gives a fully connected DAG. The main purpose of W matrix therefore is to disable the edges. Continuous W requires threshold to properly disable the edges, since it is hard for a continuous matrix to learn exactly 0 during the optimization; (ii) $ReLU(grad \pmb{p})$ already contains continuous values, and $W$ matrix also contains the continuous values. Thus, learning of the edge weights and DAG structure are not explicitly separated, resulting in complicated non-convex optimizations (see discussion below Eq.3 in [1]). With binary matrices (e.g. replacing with $Step$ function and binary $W$), we only focus on learning the graph structure, which significantly simplifies the optimization complexity.
>
> 2. **Reported failure of no-curl**: In the original No-curl work [1], they reported that direct optimizing the $W\cdot ReLU(grad \pmb{p})$ results in poor graph discovery performance (refer to rand init and rand p in Table 1 [1]). That is exactly the reason they propose to use NoTears as the initialisation and design complicated projection scheme to project initialisation to a DAG space. On the other hand, our approach does not require any projections and sampling with our modified objective can directly lead to a good graph discovery performance.
>
> 3. **Comparison of alternative approach**: To the best of our knowledge, our work is the only one that shares some similarities with No-Curl. It is important to note that our method is focused on a Bayesian causal discovery framework, while No-Curl is not explicitly designed for this purpose. Thus, No-Curl might not serve as the most appropriate baseline for our study. We further observed that No-Curl demonstrated very poor performance when directly optimized, as shown in Table 1 of [1]. This observation led us to reasonably assume that our method would outperform it. Regarding the No-Curl with the projection steps, it was found to have similar performance to No-Tears, as illustrated in Figure 1 of [1], but it is hard to adapt the projection step for Bayesian causal discovery. Dibs [2], a Bayesian causal discovery algorithm inspired by No-Tears, serves as a better comparison for our work. In most cases, our method exhibits superior performance compared to Dibs, further solidifying the effectiveness of our proposed approach.
>
> 4. **Access to code**: We have provided a link to AC containing the code and will open source it on acceptance.
>
> 5. **DAGness with approximate gradient**: Although the Sinkhorn is an approximation method, and can only output bi-stochastic matrix. To obtain a valid permutation matrix, we use Hungarian matching algorithm to ensure a valid permutation can always be obtained from this bi-stochastic matrix (see line 167). With a valid permutation matrix, a DAG structure can be guaranteed.
>
>
> We would once again like to express our gratitude to the reviewers for their valuable feedback and hope that these clarifications have effectively addressed your concerns.
>
>
> [1] Yu, Yue, et al. "DAGs with no curl: An efficient DAG structure learning approach." International Conference on Machine Learning. PMLR, 2021.
>
> [2] Lorch, Lars, et al. "Dibs: Differentiable bayesian structure learning." Advances in Neural Information Processing Systems 34 (2021): 24111-24123.

---

> > ### Comment · Area_Chair_CAFj · 2023-08-15
> > **Reviewer R7H9: Continuous vs binary W and other issues**
> >
> > Dear Reviewer R7H9,
> >
> > Have the authors addressed these issues raised in your review and does this change your assessment of the paper?

---

> > > ### Author Response · Authors · 2023-08-18
> > > **Further Questions**
> > >
> > > Dear Reviewer,
> > >
> > > Thanks for your review. Note that we have made access to the code through the AC (which will be open sourced on acceptance) and also addressed your other concerns. If there are further questions, we are happy to answer them as well. If there are no outstanding questions and if we have addressed all your concerns, given that your comments overall seem to be positive regarding our work, we would appreciate if you could consider increasing your score.

---

### Official Review · Reviewer_zwvL · 2023-07-05

**Soundness:** 3 good
**Presentation:** 3 good
**Contribution:** 3 good
**Rating:** 6
**Confidence:** 3

**Summary:**

The authors propose a method for the posterior inference of DAG structure *and* function parameters with potential applicability to arbitrary functional relations between nodes. The authors modify a novel characterization of DAGs, and interpret this characterization in terms of a sorting operation which can be relaxed to allow differentiability. The authors define priors on DAGs (in the alternative space) and function parameters and based on a specific model choice characterize likelihood. They use the resulting joint distribution to iteratively sample some parameters and conduct variational inference re. others. The authors examine the performance of their proposed methodology on various synthetic and real datasets.

**Strengths:**

- The paper is very well written. It presents the previous work, motivation for current research, and reasoning behind methodological choices very clearly.
- The paper utilizes recent, previous research intelligently and presents concrete innovations to solve well-defined problems.
- Posterior inference in the DAG structure and parameter space without some of the limitations of previous work is valuable and is likely to inspire future work.

**Weaknesses:**

- DAG model selection results have causal implications given specific model assumptions regarding generative model of the data. ANM is such a model assumption. However, it is unclear whether the identifiability results still apply in this case, given the priors defined on DAG structure and function parameters. I think the authors' work still would be valuable as only a DAG inference method; however, since the authors present their proposal as a causal discovery + inference method, this point needs further discussion.
- I think the authors' presentation should be modified to make sure their inference method is more clearly understood. Given their initial presentation, including "posterior sampling" in the title, and frequent reference to Gibbs sampling throughout the text, leads the reader think that the authors will present results with a correct MCMC algorithm and produce a full posterior distribution. However, most promising results presented by authors include their iterative algorithm that samples from the posterior of some parameters and uses variational inference for others. This is fine as a methodological choice, but their presentation leads the reader to have higher expectations, which can become crucial depending on the use case of the reader.
- Causal sufficiency assumption prevents using the current method in problems where unobserved confounding is likely. In my opinion this is acceptable given the difficulty of the problem.

**Questions:**

- Are there any potential difficulties with using the authors' method with other SCM model assumptions / likelihoods?
- What are grounds for baseline selection in experiments? I think this would be an important addition to the final text.
- In 6.4, were there model misspecification in other methods as well?

**Limitations:**

I think the authors adequately address the limitations of their work overall, however see Weaknesses section above for some important caveats.

---

> ### Author Rebuttal · Authors · 2023-08-09
>
> Thanks for your constructive feedback for our paper. We will address your concerns in the following:
>
> 1. **Identifiability**:  We would like to clarify that identifiability is a property of the SCM (the specific parametric form thereof), and not that of an estimation/inference method, which is what BayesDAG is about. With small amount of data, the prior can affect the search procedure, but it won't affect the identifiability since our priors over graphs are non-zero for any DAG. Namely, for each sampled graph and function parameters, the resulting ANM is identifiable, apart from some exceptions (e.g. linear Gaussian, etc.). With enough data (e.g.infinite data), the prior can be negligible and it will also not affect the search procedure (see Theorem 1 in [1]). Since our method can also perform causal inference task, our model is also causal inference identifiable. For each DAG and function parameters pair, we have a corresponding SEM, which the causal inference quantity can be obtained by manipulating the SEM.
>
> 2. **Clarity of the inference method**: Sorry about the ambiguity. But we want to emphasise that although the fully SG-MCMC approach does not give promising empirical results compared to the SG-MCMC+VI, proposing such framework itself is a contribution and requires further investigations regarding its inferior performances. We will make the presentation of the inference method clearer in the revised paper.
>
> 3. **Causal sufficiency**: We agree that causal sufficiency is a strong assumption in practice, but under the scope of this paper, we only consider the system without latent confounders. This is also a common assumption adopted by all our baselines and most previous work. Future work is needed to relax this constraint.
>
> 4. **Potential difficulty with other SCM**: It depends on which SCM has been chosen. The core assumption required by our method is that the SCM should be structurally identifiable. For example, post-nonlinear model, which is identifiable, can be used to replace the ANM model in our case.
>
> 5. **Baseline selection**: The selection criteria for baselines are the following: (1) It should be a Bayesian causal discovery method; (2) the baselines should cover both linear (BCD nets) and non-linear models (Dibs); (3) the baseline should have quasi-Bayesian method adapted from traditional causal discovery approach (BGES). These principles ensure a comprehensive set of baselines to demonstrate the effectiveness of our method.
>
> 6. **Model misspecification in 6.4**: Yes, our method, along with other baselines, should have model misspecification, since the ground truth mechanism may not have additive noise structure.
>
> We would once again like to express our gratitude to the reviewers for their valuable feedback and hope that these clarifications have effectively addressed your concerns.
>
>
> [1] Geffner, Tomas, et al. "Deep end-to-end causal inference." arXiv preprint arXiv:2202.02195 (2022).

---

> > ### Comment · Area_Chair_CAFj · 2023-08-15
> > **Reviewer zwvL: Have the authors addressed your concerns?**
> >
> > Dear Reviewer zwvL,
> >
> > Have the authors addressed the potential weaknesses raised in your review and your questions? Does their rebuttal change your assessment of the paper?

---

> > > ### Comment · Reviewer_zwvL · 2023-08-15
> > > **Thanks for the comments**
> > >
> > > I thank the authors for their comments and clarifications. I appreciate the clarification on the identifiability problem, baseline selection, and application to other SCM's. As I said in my review, causal sufficiency assumption is tolerable given other developments in the paper.
> > >
> > > To clarify my comments about MCMC: I did not mean to say that the SG-MCMC procedure was not a contribution. However, its performance is inferior to the extent that its results were deferred to the Appendix and SG-MCMC-VI results were used instead. I do not have a problem with that per se. My comment was about the abstract or the title not mentioning this at all. An uniformed reading of the abstract makes the reader think that the effective results mentioned in the abstract are obtained by SG-MCMC. MCMC methods and VI methods (or hybrids of the two) are categorically different from a Bayesian inference perspective, and I feel these lines are blurred in an unwarranted fashion not only in the abstract but throughout the text e.g. in L176 where the authors mention combining MCMC and VI "in a Gibbs sampling manner". I keep my score as is, and recommend that the authors make sure this distinction is emphasized clearly where relevant. For example, the title could be changed to say: "BayesDAG: Gradient-Based Posterior Inference for Causal Discovery".

---

> > > > ### Author Response · Authors · 2023-08-16
> > > >
> > > > Thanks a lot for your response.
> > > >
> > > > Thanks a lot for clarification regarding MCMC. Our point of "sampling" in the title and abstract was to emphasize that, unlike prior methods, which exclusively use either VI or MH based MCMC, we can use gradient based sampling procedure for Bayesian Causal Discovery for all parameters (except for W, which we rely on VI). However, based on your comment, we will make sure the distinction is more empasized and will make the following changes:
> > > >
> > > > 1. Line 9 " _In this work, we introduce a scalable Bayesian causal discovery framework based on combination of stochastic gradient Markov Chain Monte Carlo (SG-MCMC) and Variational Inference (VI) that overcomes these limitations._"
> > > > 2. As per your suggestion, change the title to "BayesDAG: Gradient-Based Posterior Inference for Causal Discovery", if such a change is allowed after acceptance.
> > > >
> > > > We are happy to answer any further questions/ concerns, if any, you may have.

---

> > > > > ### Comment · Reviewer_zwvL · 2023-08-19
> > > > > **Thanks**
> > > > >
> > > > > I appreciate the authors' proposed changes. I think the paper body (esp. introduction) needs a similar revision pass for conceptual clarity, but this is something that can be feasibly completed until camera-ready (should the paper be accepted).

---

> > > > > > ### Author Response · Authors · 2023-08-21
> > > > > >
> > > > > > Thanks a lot for your comment. We will make sure to emphasize the difference between SG-MCMC and VI for W in introduction as well for the revision.

---

### Official Review · Reviewer_3g7c · 2023-07-06

**Soundness:** 3 good
**Presentation:** 3 good
**Contribution:** 3 good
**Rating:** 7
**Confidence:** 3

**Summary:**

The paper proposes a Bayesian causal discovery method based on a novel parametrization of the binary DAG space and SG-MCMC. The proposed method does not rely on DAG regularization nor restricted to linear models, overcoming the limitations of prior approaches. Experimental results demonstrate the competitive performance of the proposed method compared to existing approaches.


**Strengths:**

The paper is well-written and easy to understand. It is easy to follow the core idea of the proposed method.

The proposed method is sound and well-motivated (i.e., there are apparent limitations of previous work but this method overcomes such issues.)

Several techniques are employed smoothly to propose the method.

Experimental results demonstrate the effectiveness and scalability of the proposed method.


**Weaknesses:**

For the empirical evaluation, comparison with MCMC approach [1] is missing. Also, AUROC is not reported, which is widely used for the evaluation of uncertainty quantification in Bayesian causal discovery literature.

The proposed method is claimed to be scalable and computational complexity is analyzed, but the actual computation cost (e.g., wall clock time) is not compared with DiBS. Computation resources they used are also not provided (e.g., CPU, GPU).

Similarly, “per node degree 2” seems very limiting which results in a very sparse graph for large d. Large d will certainly affects the performance of likelihood computation and posterior sampling of W where more edges may demonstrate dependencies.

[1] Improving markov chain monte carlo model search for data mining, 2003


**Questions:**

Details of the weight-sharing mechanism (line 186) are missing. While the size of the networks is very small, how does it impact the performance of the proposed method other than reducing the total network parameters?

Typo: (Line 230) focusonly


**Limitations:**

Some of the assumptions can be viewed as limitations but I am fine with those (they are crucial to yields the proposed methods)

---

> ### Author Rebuttal · Authors · 2023-08-09
>
> Thanks for your constructive feedback for our paper. We will address the concerns in the following:
>
> 1. **Missing MCMC baselines**: We acknowledge the importance of a comprehensive analysis. Note that all MCMC methods are only defined on linear models, usually where the parameters can be marginalized out. They are also not scalable, limiting the settings which we can compare it with. There has been extensive research showing their lack of scalability, mixing and convergence speed (see related work). In addition, it has been shown that DIBS [1] outperforms all MCMC methods (refer to Figure 2, 3, 4, and Table 1 in [1]). The superior performance of our method, as compared to DIBS, led us to believe that MCMC as a baseline might not be the best choice. However, given your comment, we will include a comparison to an MCMC baseline for the 5 variable setting for camera-ready, even though we believe adding it will not change our conclusion.
>
> 2. **Missing AUROC metric**: Our primary goal was to select a diverse set of metrics that provide a comprehensive assessment of the performance, capturing different aspects of the model. While AUROC is a metric for evaluating graph quality, we believe that its utility in our case is somewhat limited, as it shares similarities with the SHD and F1 score. These latter metrics, which are incorporated into our evaluation, also consider the inferred graph's quality, arguably making the inclusion of AUROC somewhat redundant for our purposes. Apart from the quality of graph posterior, posterior over function parameters is also crucial for Bayesian causal discovery. This aspect is not captured by the AUROC metric. To account for this crucial facet, we opted for the held-out likelihood as an evaluation metric. This choice is grounded in its widespread use in the literature as a reliable indicator of Bayesian inference quality [2,3]. Meanwhile, we can include the AUROC in the camera-read version.
>
> 3. **Computational efficiency**: We have included the wall-clock comparison and generalization to $d>100$ datasets in the PDF. It is important to emphasize that our approach is capable of handling significantly larger dataset dimensions, scaling up to $d>100$ cases, and achieve faster convergence compared to BCD and Dibs in terms of wall-clock time under all dimensionalities. On the other hand, DIBS is limited to $d\leq50$ with a single GPU, and BCD is limited to $d<150$.Moreover, our paper includes an in-depth computational complexity analysis (refer to lines 232-241). This analysis further strengthens the claim that our approach exhibits a competitive edge over DIBS.
>
> 4. **Per node degree of 2**: Our choice is motivated by its prevalence in the literature, particularly when it comes to synthetic ER and SF datasets [4]. Notably, DIBS [1] also employed this choice for their experiments. By adopting this, we ensure that our experiments are consistent with established benchmarks, thereby facilitating a more meaningful comparison between our method and baselines.
>
> 5. **Weight-sharing mechanism and size of networks**: Sorry for the ambiguity. By formulation of Eq.10, each node requires two separate neural networks. Thus, in total, it requires $2d$ number of different networks to train, incurring high computational cost. To avoid that, instead, we have a separate trainable embedding $\vu_i$ for each node. Therefore, we only need two neural networks with this trainable embedding to differentiable different nodes. This is equivalent to sharing the weights across the $2d$ neural networks [5]. also adopted the same sharing mechanism. We will add a clearer explanation in the revised paper. The reason we choose this network size is that it is already a common choice in the literature [1,5]. In fact, Dibs [1] used even smaller network sizes (2 hidden layers with 5 units). Our method can be easily extended to larger network sizes and it shouldn't impact the performance much, since SG-MCMC is designed to accommodate the large network and dataset sizes [6].
>
> We would once again like to express our gratitude to the reviewers for their valuable feedback and hope that these clarifications have effectively addressed your concerns.
>
> [1] Lorch, Lars, et al. "Dibs: Differentiable bayesian structure learning." Advances in Neural Information Processing Systems 34 (2021): 24111-24123.
>
> [2] Gong, Wenbo, Yingzhen Li, and José Miguel Hernández-Lobato. "Meta-learning for stochastic gradient MCMC." arXiv preprint arXiv:1806.04522 (2018).
>
> [3] Lorch, Lars, et al. "Amortized inference for causal structure learning." Advances in Neural Information Processing Systems 35 (2022): 13104-13118.
>
> [4] Zheng, Xun, et al. "Dags with no tears: Continuous optimization for structure learning." Advances in neural information processing systems 31 (2018).
>
> [5] Geffner, Tomas, et al. "Deep end-to-end causal inference." arXiv preprint arXiv:2202.02195 (2022).
>
> [6] Chen, Changyou, et al. "Bridging the gap between stochastic gradient MCMC and stochastic optimization." Artificial Intelligence and Statistics. PMLR, 2016.

---

> > ### Comment · Reviewer_3g7c · 2023-08-12
> >
> > Thank you for your response. The authors answers perfectly cleared up almost all of my concerns.
> >
> > Regarding 4, "per node degree of 2", it is understandable since prior work also did the same. However, setting degree to 3 is not a difficult request, and it is good to see how performance drops as we increase the degree.
> >
> > Regardless, I would like to keep my score ("accept").

---

> > > ### Author Response · Authors · 2023-08-16
> > >
> > > Thanks a lot for your response and confirming your positive score. We are happy to hear that our response cleared up your concerns.
> > >
> > > Regarding experiments with per node degree 3, given the limited time and resources for the rebuttal, we prioritized scaling experiments (see rebuttal pdf) instead of per node degree 3, as we believe the experiments and conclusions would be very similar, if not the same, as compared to the experiments with per node degree 2.  In order to further address your issue and as a response to R7H9, we have provided the source code which can easily handle running for per node degree 3. We will open source this code on acceptance.
> > >
> > > Also note that the real world/ semi-synthetic datasets contain graphs which are not necessarily per node degree 2, where our approach performs well.
> > >
> > > We are happy to answer any further questions you may have.

---

### Official Review · Reviewer_hANm · 2023-07-07

**Soundness:** 3 good
**Presentation:** 3 good
**Contribution:** 2 fair
**Rating:** 5
**Confidence:** 4

**Summary:**

The paper proposes a novel Bayesian causal discovery (BCD) method that infer the posterior distribution $p(G|\mathcal{D})$ by projecting the DAG $G$ into an equivalent search space. Instead of sampling $G$, the method constructs the posterior distribution by sampling a binary matrix $W$ and potential vector $p$ with via MCMC sampling and variational inference. The Bayesian causal discovery method can scale up to 100 variables and achieves better accuracy on large datasets.

**Strengths:**

- The idea of employing the projection framework from DAG-Nocurl paper is interesting. Especially, the difficulties of the sampling based posterior distribution estimation for DAG learning methods lie in the order of parents sampling. The potential function p automatically reserves the causal order.

- The proposed method is a combination of sampling-based method and variational inference method. Compared to the state-of-art BCD methods that adopt VI, the proposed method achieves better SHD, especially on high-dimensional data.

**Weaknesses:**

- (**Major**) The experiments are not comprehensive. There is a trade-off between efficiency and accuracy compared sampling-based approach to VI approach. Compared to the existing VI-based BCD methods, It is possible that the proposed approaches are more accurate but also suffer from low efficiency. Please refer to the question section for details.

- (**Minor**) The tuning of hyperparameters such as scale of p and theta. Since the original framework of DAG-Nocurl is derived for continuous parameterization, the algorithm requires the tuning of additional hyperparameters, which increases the training difficulty. But I understand this is a minor concern.

**Questions:**

My question focus on the experiment section:

- It seems that the BayesDAG achieves better accuracy on cases $d>=70$. I wonder if authors can also show the efficiency (as in runtime) of different methods? I suspect that BayesDAG would take longer to converge than the VI-based approaches due to the Gibbs sampling procedure. The strength of BayesDAG is better demonstrated if it can achieve much better accuracy with a slight compromise on the efficiency.
- Some synthetic data is generated based on the unidentifiable linear SEM with non-equal variances. How does the identifiability of the employed SEM affect the BayesDAG method (How does the non-equal variances assumption influence the Eq. (11))?
- The paper only show the empirical results on $d\leq 100$ variables. I am wondering if authors can show BayesDAG can scale up to data with higher dimensions. The gradient-based causal discovery methods such as GraN-DAG can also scale up to 100 variables and finish within reasonable runtime with the help of GPU.
- It would be better if the authors can compare to traditional scalable causal discovery methods such as FGES?

**Limitations:**

The idea of the paper seems interesting and the theory is sound. I concern that the Gibbs sampling procedure in the propose algorithm may compromise its efficiency and scalability.

---

> ### Author Rebuttal · Authors · 2023-08-09
>
> Thanks for your constructive feedback for our paper. We will try to address the raised concern in the following.
>
> 1. **Major concern**: We have included a wall-clock time comparison in the supplementary PDF. From the comparison, we can see our approach, compared to Dibs and BCD, converges faster while obtaining better performance in most cases. DDS, though converges faster, performs significantly worse than our approach. In addition, it is important to note that the BCD net is a **linear method**, while our approach is **fully non-linear**. However, this simplicity in model assumption may be too restrictive for certain problems, limiting the potential for accurate causal discovery. In contrast, our method not only demonstrates performance advantages for dimensions $d > 70$, but also outperforms competing methods, including BCD net, for $d = 30$ and $d = 50$, along with the faster convergence speed. This can be clearly observed in Figure 3 of our paper. These results serve as strong evidence for the benefits of adopting a non-linear model in Bayesian causal discovery.
>
> 2. **Minor concern**: In response to the reviewer's concerns about hyper-parameter tuning, we have provided a comprehensive list of hyper-parameters used in our experiments within Appendix D. To address the challenge of tuning additional hyperparameters, we have conducted ablation studies presented in Appendix E.3. These studies examine the sensitivity of the initialized $\pmb{p}$, the number of MCMC chains used, and the sampler noise scale for $\pmb{p}$ and $\pmb{\Theta}$. These ablation studies offer valuable guidelines for tuning the additional hyperparameters. To further assist the readers, we will include a concise paragraph in the revised paper, explaining how to select these hyperparameters.
>
> 3. **Identifiability**: It is important to emphasize that identifiability is an inherent property of the model (and not that of the inference/estimation method, which is what BayesDAG is about), which we use additive noise model to ensure this. With small amount of data, the prior term can affect the search procedure, but it will not affect the identifiability since our prior will put non-zero probability on any DAG and function parameters. On the other hand, with enough data (e.g.~infinite data), the prior term can be negligible, and will also not affect our search procedure (see Theorem 1 in [1]). In the case of experiment 6.1.1, the ground truth data generative mechanism is not identifiable, and only identifiable up to Markov equivalence class; however, this does not affect our inference procedure in any sense. One advantage of our method is that it can recover multiple graphs that fit the data equally well, which can be used to test the model's uncertainty estimation quality.
>
> 4. **Higher dimensional problem**: As requested, we have included the performance in even higher dimensions in the supplementary material (with 40GB A100 GPU). We are the only method that is capable of generalising to $d>100$ among the baselines apart from BGES, which is a quasi-Bayesian approach. BCD and Dibs are not scalable enough to run under this high dimensionality, which demonstrates our method is more memory-efficient compared to most of them. Additionally, we would like to clarify that a direct comparison between our method and GraN-DAG may not be entirely appropriate for the following reasons: (i) GraN-DAG is not a Bayesian causal discovery method; it focuses on inferring a single DAG, whereas our method is designed to infer a distribution over DAGs; (ii) GraN-DAG is based on continuous-relaxation, which may result in graphs that do not strictly adhere to the DAG structure. In contrast, one advantage of our method is to infer DAGs directly.
>
> 5. **FGES**: Regarding traditional causal discovery baselines, we have indeed included a comparison with the bootstrap GES method in our paper. Please refer to Figure 2, Figure 3, and Tables 1 and 2 for detailed comparisons. It is important to note that Fast GES is a variant of GES, but it is not a Bayesian causal discovery method, and only infer a single graph.
>
> We would once again like to express our gratitude to the reviewers for their valuable feedback and hope that these clarifications have effectively addressed your concerns.
>
> [1] Geffner, Tomas, et al. "Deep end-to-end causal inference." arXiv preprint arXiv:2202.02195 (2022).

---

> > ### Comment · Area_Chair_CAFj · 2023-08-15
> > **Reviewer hANm: Have the authors addressed your concerns regarding the experiments?**
> >
> > Dear Reviewer hANm,
> >
> > The authors have replied to your concerns regarding the experiments and also added new results in their global reply's pdf. Could you please have a look and let us know if this changes your original assessment?

---

> > > ### Author Response · Authors · 2023-08-18
> > >
> > > Dear Reviewer,
> > >
> > > Thanks a lot for your review. We have provided detailed scaling analysis and walltime results as per your concerns. If there are any outstanding questions, we are happy to answer them as well. If there are no further questions and our response addressed your concerns, we would be happy if you could consider increasing your score.

---

> > ### Comment · Reviewer_hANm · 2023-08-21
> >
> > Thanks the reviewers for the responses.
> >
> > I believe the reviewers have addressed my concerns.
> >
> > Overall, I believe this paper is interesting and I would like to keep my score as borderline accept.

---

### Author Rebuttal · Authors · 2023-08-09

We would like to express our sincere appreciation to all reviewers for their valuable time and efforts in providing constructive feedbacks for our paper. We are delighted to hear that the reviewers generally find our work interesting (hANm, R7H9), sound and well-motivated (3g7c, zwvL), well-written (3g7c, zwvL, R7H9), valuable and is likely to inspire future work  (zwvL).

We have taken into consideration the common concerns raised, specifically regarding computational efficiency and scalability. In response, we have included a comprehensive comparison of wall-clock times and performance results for dimensions $d > 100$ in the supplementary PDF. In summary, we would like to emphasize that BayesDAG, as a nonlinear Bayesian approach, demonstrates better computational efficiency in comparison to existing state-of-the-art Bayesian causal discovery algorithms, such as Dibs and BCD nets. It is also worth noting that algorithms like Dibs and DDS are unable to run for dimensions $d > 70$ within a single GPU setup, and we are the only one that is capable of generalising to $d>100$ cases (with a single 40GB A100 GPU). These additional experiments effectively demonstrate the computational advantages of our approach compared to the baselines. In applications like gene regulatory network inference, approaches that can deal with several hundred variables are required, and our contribution takes a positive step in this direction. Just like NoTears, our work, based on NoCurl, and our theoretical contributions lay ground work for scalable causal discovery approaches in more complex settings, for example with hidden variables. We also shared our code as requested by R7H9.

---

### Decision · Program_Chairs · 2023-09-21

**Decision:**

Accept (poster)

**Comment:**

This paper proposes a Bayesian approach to directly acyclic graph (DAG) structure learning based on a hybrid SG-MCMC sampling and variational inference scheme for drawing samples from the DAG posterior. This work is related to the NoCurl approach of Yu et al., where the space of DAGs is represented in the space of a skew-symmetric matrix W and a potential vector p. While NoCurl provides a point-estimate, the proposed method focuses on the Bayesian setting (and hence providing a full posterior over DAGs).

I thank the reviewers and the authors for their discussions. I believe all the reviewers are positive about this paper in that it represents a contribution to the problem of DAG estimation (Bayesian and non-linear cases). Most concerns have been addressed by the authors in the rebuttal. In particular, that of considering computational efficiency vs. accuracy.